# CONTRASTIVE REPRESENTATION REGULARIZATION FOR VISION-LANGUAGE-ACTION MODELS

## ABSTRACT

Vision-Language-Action (VLA) models have shown its capabilities in robot manipulation by leveraging rich representations from pre-trained Vision-Language Models (VLMs). However, their representations arguably remain suboptimal, lacking sensitivity to robotic signals such as control actions and proprioceptive states. To address the issue, we introduce *Robot State-aware Contrastive Loss (RS-CL)*, a simple and effective representation regularization for VLA models, designed to bridge the gap between VLM representations and robotic signals. In particular, RS-CL aligns the representations more closely with the robot's proprioceptive states, by using relative distances between the states as soft supervision. Complementing the original action prediction objective, RS-CL effectively enhances control-relevant representation learning, while being lightweight and fully compatible with standard VLA training pipeline. Our empirical results demonstrate that RS-CL substantially improves the manipulation performance of state-of-the-art VLA models; it pushes the prior art from 30.8% to 41.5% on pick-and-place tasks in RoboCasa-Kitchen, through more accurate positioning during grasping and placing, and boosts success rates from 45.0% to 58.3% on challenging real-robot manipulation tasks.

## 1 INTRODUCTION

Vision-Language-Action (VLA; Zitkovich et al. 2023) models have emerged as a powerful framework for robot manipulation, leveraging pre-trained Vision-Language Models (VLM; Liu et al. 2023b) to provide rich visual and semantic grounding for control policies. Among the state-of-the-art VLA models, the common design is to employ a generative action decoder conditioned on VLM-derived representations (Black et al., 2025b; Bjorck et al., 2025). These decoders are trained with an action prediction loss, supervised by the ground-truth sequence of actions.

Prior studies have shown that fine-tuning the VLM alongside training the action decoder is essential to the action prediction performance of VLA models. This is because VLM representations are typically trained on large-scale visual instruction datasets, but have not been explicitly exposed to robotic modalities, such as low-level control actions and proprioceptive information. Consequently, training VLA models conditioned on frozen VLM representations leads to suboptimal performance, as the VLM lacks the capability to capture robotic signals (Driess et al., 2025).

Many recent works have proposed different approaches to train the VLM backbone in VLA models to tackle this issue. A widely adopted strategy is to directly update the VLM via gradients from the action prediction objective (Black et al., 2025b; Bjorck et al., 2025). Beyond this, several works introduce auxiliary objectives, such as jointly training the VLM backbone with curated instruction datasets (Yang et al., 2025), or blocking gradients from the action decoder instead learning to generate intermediate subtasks and discretized actions (Driess et al., 2025). Another line of work further trains the VLM on embodied reasoning or spatial grounding tasks using robotics datasets (Ji et al., 2025; Luo et al., 2025; Azzolini et al., 2025; GEAR, 2025), or autoregressively predicts discretized actions (Kim et al., 2025; Black et al., 2025a) before fine-tuning them for continuous action prediction. While these approaches help bridge the gap between general-purpose VLM representations and the demands of action prediction, they often require additional training stages or carefully curated datasets.

In contrast, we aim to directly refine VLM representations to better serve action generation, while remaining efficient and seamlessly compatible with the existing VLA training pipelines. In particular, we focus on contrastive learning, as it provides a principled way to refine representations by defining

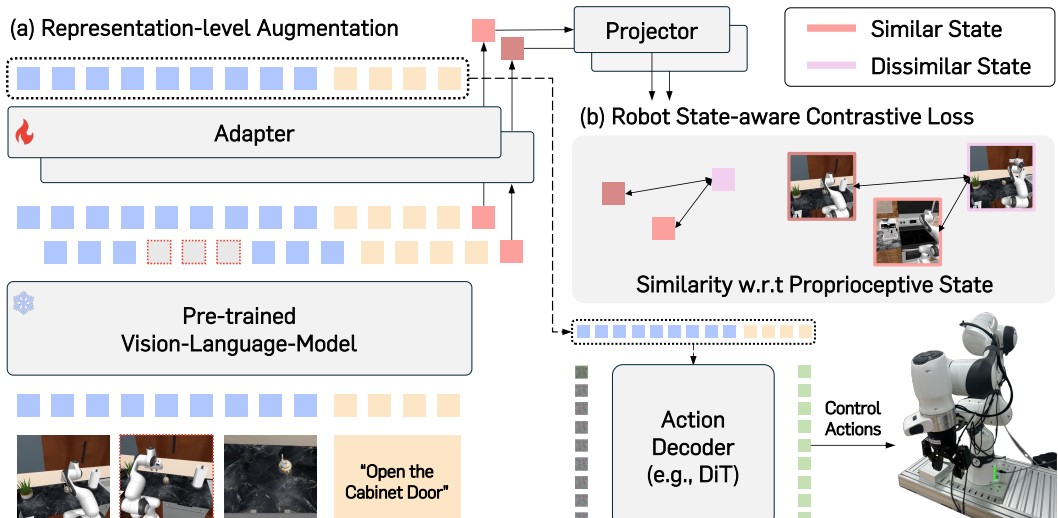

Figure 1: **Overview.** We extend the standard VLA model training framework with a contrastive regularization path. Embeddings from the pre-trained VLM are augmented by the *view cutoff* operation applied on the feature slice corresponding to a randomly selected observation view, and are optimized with our *Robot State-aware Contrastive Loss* to attract samples with similar proprioceptive states, complementing the action prediction loss.

similar and dissimilar pairs, effectively structuring the embedding space. The specific choice of pair construction determines what the embeddings should capture, ranging from semantic relations between modalities (Radford et al., 2021) to temporal dynamics and policy-relevant representations (Sermanet et al., 2018; Nair et al., 2022; Ma et al., 2023). Inspired by this perspective, we introduce a contrastive objective that explicitly guides the representations to capture robotic signals, in particular the robot's proprioceptive states. By jointly optimizing the VLM representation with the standard action prediction loss, we forge representations that are not only semantically rich but also deeply grounded in the robot's physical state, leading to accurate action prediction.

**Contribution.** In this paper, we introduce a novel self-supervised regularization objective for VLA models, termed *Robot State-aware Contrastive Loss (RS-CL)*, a loss that explicitly shapes VLM representations toward capturing robotic signals. Different from the conventional contrastive loss, RS-CL assigns pairwise weights based on the distances between robot proprioceptive states, guiding the representations to better reflect robot control-relevant structure. In addition, we propose an representation-level augmentation for VLA models, called *view cutoff*. This augmentation constructs alternative embeddings by masking out the feature corresponding to a randomly selected observation view. By operating at the representation-level and minimizing the forwarding process through the pre-trained VLM, RS-CL remains lightweight and fully compatible with existing training pipeline.

We extensively evaluate the effectiveness of RS-CL under manipulation benchmarks such as RoboCasa-Kitchen (Nasiriany et al., 2024) and LIBERO (Liu et al., 2023a). For instance, RS-CL pushes the prior art VLA model from 48.2% to 53.0% (+4.8%), 63.9% to 67.2% (+3.3%), and 65.7% to 69.7% (+4.0%) on RoboCasa-Kitchen, with 30, 100, and 300 demonstrations, respectively. We emphasize that RS-CL gives larger improvement of 30.3% to 41.5% (+11.2%) on pick-and-place tasks, which requires precise positioning during grasping and placing. Finally, we show that RS-CL is applicable to real-robot hardware experiments, showing improvement from 45.0% to 58.3% (+13.3%) on challenging manipulation tasks.

In summary, our contributions are as follows:

- We introduce *Robot State-aware Contrastive Loss (RS-CL)*, a novel objective for VLA models that explicitly aligns VLM representations with proprioceptive states.
- We design RS-CL to operate directly at the representation alongside the original action prediction objective. Therefore RS-CL remains lightweight and compatible with the existing training pipeline.
- We validate RS-CL across diverse training scenarios on manipulation benchmarks and real-world experiments, showing consistent improvements over the state-of-the-art VLA models.

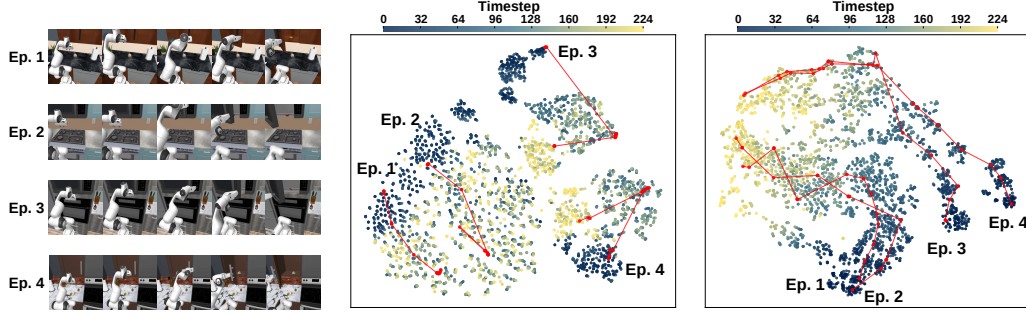

(a) Task trajectories.  (b) Pre-trained VLM representations.  (c) RS-CL aligned representations.

Figure 2: **Training VLM representations for action prediction. (a)** We visualize VLM embeddings of robot episodes performing the same task "Open the microwave / cabinet door" across different scenes in RoboCasa-Kitchen. **(b)** Pre-trained VLM representations are dominated by the visual appearance (*e.g.*, distractor objects). **(c)** RS-CL guides embeddings to align with the robot's proprioceptive states, yielding representations that capture common robotic signals (*e.g.*, the robot's current pose, next control action) across environments, therefore aligning all episodes by the task progress.

## 2 METHOD

In this section, we introduce *Robot State-aware Contrastive Loss (RS-CL)*, which enhances the action prediction capability of VLA models by guiding the representation to capture low-level robotic signals, particularly the proprioceptive states. We describe the VLA training framework in Sec. 2.1 and present our proposed method, RS-CL, in Sec. 2.2. An overview of our method is shown in Fig. 1.

### 2.1 VISION-LANGUAGE-ACTION MODEL

VLA models are trained to predict the next action chunk $\mathbf{A}_t = [\mathbf{a}_t, \mathbf{a}_{t+1}, \ldots, \mathbf{a}_{t+H}]$ of horizon $H$ at current timestep $t$, from a set of observation images from $V$ different views $\mathbf{O}_t^V = \{\mathbf{o}_t^1, \mathbf{o}_t^2, \ldots, \mathbf{o}_t^V\}$, a task instruction $\mathbf{c}$, and the robot's proprioceptive state $\mathbf{q}$. A standard framework for VLA models (Black et al., 2025b; Bjorck et al., 2025) encodes multimodal inputs $[\mathbf{O}_t^V, \mathbf{c}]$ using a pre-trained VLM into a hidden representation, and pass it to the action decoder. In practice, we train a lightweight adapter module $f_\phi$ upon the VLM and freeze the VLM, following GEAR (2025). $f_\phi$ processes the output of the VLM as $\mathbf{h} = f_\phi\big(\text{VLM}(\mathbf{O}_t^V, \mathbf{c})\big) \in \mathbb{R}^{N \times d_{\text{model}}}$, where $N$ is the number of input tokens for the VLM and $d_{\text{model}}$ is the size of the hidden dimension.

An action decoder $D_\theta$ generates $\mathbf{A}_t$ conditioned on $\mathbf{h}$ with the current robot state $\mathbf{q}$. Similar to prior works (Black et al., 2025b; Bjorck et al., 2025), we adopt the DiT (Peebles & Xie, 2023) architecture for the $D_\theta$ and train with the flow-matching objective (Lipman et al., 2023; Liu, 2022):

$$\mathcal{L}_{\text{FM}}(\theta, \phi) = \mathbb{E}_s \left[ \|D_\theta(\mathbf{h}, \mathbf{A}_t^s, \mathbf{q}) - (\epsilon - \mathbf{A}_t)\|_2^2 \right], \tag{1}$$

where $\mathbf{A}_t^s = s\mathbf{A}_t + (1-s)\epsilon$ is an interpolated action chunk at the flow-matching timestep $s \in [0, 1]$ sampled from a prior distribution $p(s)$. After training, $D_\theta$ generates $\mathbf{A}_t$ through an iterative denoising process starting from a random Gaussian noise $\epsilon \sim \mathcal{N}(\mathbf{0}, \mathbf{I})$.

### 2.2 ROBOT STATE-AWARE CONTRASTIVE LOSS

While VLMs acquire rich semantic representations from Internet-scale vision–language data, they lack exposure to robotic modalities such as low-level control actions and proprioceptive states. As a result, their embeddings are strongly shaped by the visual appearance and often fail to capture signals relevant to robot control. This misalignment is evident when we visualize the VLM embeddings of robot trajectories for the same manipulation task (*e.g.*, Open the microwave / cabinet) across different environments in RoboCasa-Kitchen (see Fig. 2a). We observe that VLM embeddings are dominated by the visual cues, such as presence of large objects or background textures (see Fig. 2b), rather than control-relevant factors like the robot's current pose or the next action needed to complete the task.

This misalignment motivates our central hypothesis: explicitly aligning VLM representations with their physical state will improve action prediction. Based on this hypothesis, we introduce *Robot State-aware Contrastive Loss (RS-CL)*, an auxiliary objective for VLAs that regularizes the VLM's representation space using supervision from the robot's proprioceptive states. Our key idea is a contrastive loss that uses the distances between proprioceptive states to assign soft weights to similarity scores, which effectively guides the representation space to be aligned with robotic signals. As an auxiliary objective, RS-CL complements the original action prediction loss, enabling the entire model to be trained end-to-end in a single stage. Concretely, RS-CL consists of three key components: a *learnable summarization token* that amortizes long VLM outputs, a *weighting scheme* for robot state supervision, and a *representation-level augmentation* strategy for lightweight representation learning.

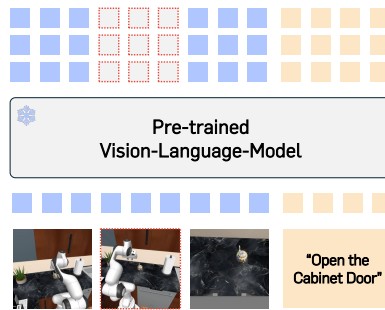

Figure 3: **Representation-level augmentation for contrastive pairs.** *View cutoff* is an simple augmentation that randomly masks out the embedding slice of one observation view from the VLM representation.

**Amortizing VLM embeddings for representation learning.** Applying contrastive learning on the full sequence of VLM embeddings $\mathbf{h} \in \mathbb{R}^{N \times d_{\text{model}}}$ is impractical as the sequence length $N$ is typically large, leading to high computational cost and diluted learning signals. To address this, we introduce a *learnable summarization token* $\mathbf{u} \in \mathbb{R}^{1 \times d_{\text{model}}}$ to produce a compact representative embedding of the sequence. Specifically, $\mathbf{u}$ is appended to the VLM output and processed by the adapter $f_\phi$:

$$[\mathbf{h}, \mathbf{w}] = f_\phi\big(\text{VLM}(\mathbf{O}_t^V, \mathbf{c}) \oplus \mathbf{u}\big), \tag{2}$$

where $\mathbf{w}$ denotes the output corresponding to the summarization token and $\oplus$ denotes concatenation along the sequence dimension. Finally, $\mathbf{w}$ is projected by a lightweight projector $g_\psi$ into $\mathbf{z} = g_\psi(\mathbf{w})$, providing a compact summary for contrastive learning (Chen et al., 2020), while the original embeddings $\mathbf{h}$ serves as the conditioning input to the action decoder.

**Incorporating robot states into contrastive learning.** To effectively restructure the VLM representation space to capture robotic signals, we introduce a supervised contrastive learning objective assigned with *soft weights* (Khosla et al., 2020; Suresh & Ong, 2021), that incorporate the distance between proprioceptive states. Conceptually, embeddings associated with similar proprioceptive states receive higher weights, are pulled closer in the representation space. We consider InfoNCE (Oord et al., 2018) for the contrastive loss, which is widely used in practice (Laskin et al., 2020; Nair et al., 2022; Ma et al., 2023). Formally, our *Robot State-aware Contrastive Loss (RS-CL)* is defined as a weighted variant of the InfoNCE loss:

$$\mathcal{L}_{\text{RS-CL}}\big(\{\mathbf{z}\}_{i=1}^B, \{\tilde{\mathbf{z}}\}_{j=1}^B; \phi, \psi\big) = -\sum_{i=1}^B \sum_{j=1}^B w_{ij} \log \frac{\exp\big(\text{sim}(\mathbf{z}_i, \tilde{\mathbf{z}}_j)/\tau\big)}{\sum_{k=1}^B \exp\big(\text{sim}(\mathbf{z}_i, \tilde{\mathbf{z}}_k)/\tau\big)}, \tag{3}$$

where $\{\tilde{\mathbf{z}}\}_{j=1}^B$ is the augmented batch of $\{\mathbf{z}\}_{i=1}^B$, sim denotes the cosine similarity, and $\tau > 0$ is a temperature that controls the sharpness of similarity. The soft weights $w_{ij}$ are computed from the relative distance between proprioceptive states $\mathbf{q}_i, \mathbf{q}_j$. In practice, we use the Euclidean distance and formulate $w_{ij}$ as follows:

$$w_{ij} = \frac{\exp(-\|\mathbf{q}_i - \mathbf{q}_j\|_2/\beta)}{\sum_{k=1}^B \exp(-\|\mathbf{q}_i - \mathbf{q}_k\|_2/\beta)}, \tag{4}$$

where $\beta > 0$ is a temperature that controls the sharpness of the mapping from distance to weight. The complete training objective integrates the proposed RS-CL with the action prediction objective, implemented as the flow-matching loss in Eq. 1:

$$\mathcal{L} = \mathcal{L}_{\text{FM}} + \lambda \mathcal{L}_{\text{RS-CL}}, \tag{5}$$

where we jointly optimize $\theta$, $\phi$, and $\psi$.

**Representation augmentation for contrastive pairs.** The primary goal of our augmentation strategy is to generate diverse contrastive pairs while preserving the semantics tied to the robot's proprioceptive states. In line with this goal, we exploit the property that VLA models commonly

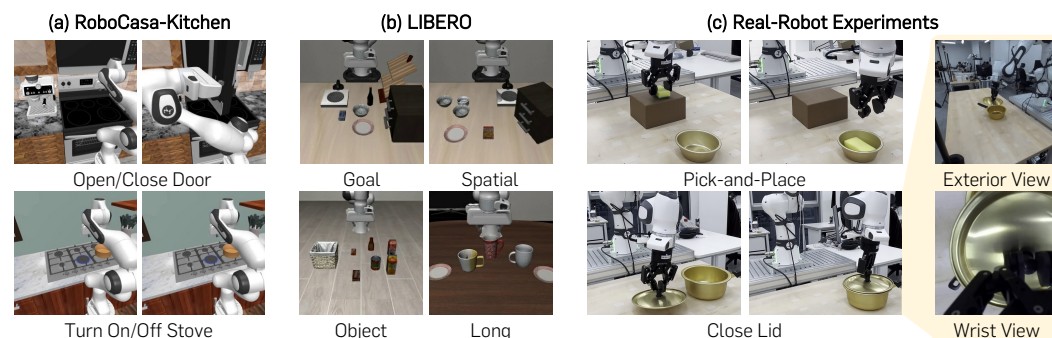

Figure 4: **Example of tasks used in our experiments.** We study RS-CL on multitask simulation benchmarks of **(a)** RoboCasa-Kitchen (Nasiriany et al., 2024) and **(b)** LIBERO (Liu et al., 2023a). In addition, we consider **(c)** real-robot manipulation tasks considering pick-and-place, and a close lid task, utilizing two camera viewpoints.

process observations of the same scene from multiple views, and propose *view cutoff* (See Fig. 3), a simple representation-level augmentation inspired by cutoff (Shen et al., 2020). *View cutoff* randomly selects a single view index $i \in \{1, \ldots, V\}$ and masks out the corresponding feature slice from the VLM output $\text{VLM}(\mathbf{O}_t^V, \mathbf{c})$. Unlike data-level augmentations requiring additional forward passes through the VLM for each augmented batch, view cutoff operates at the representation level, obtaining alternative representations with minimal overhead. As a result, only the lightweight adapter $f_\phi$ and projector $g_\psi$ are required to process the augmented variants, making the method substantially more efficient, yet still providing diverse pairs for contrastive learning.

## 3 EXPERIMENTS

In this section, we evaluate the effectiveness of RS-CL across diverse training scenarios. In Section 3.1, we examine its impact when applied on top of large-scale pre-trained state-of-the-art Vision-Language-Action (VLA) models on challenging multitask manipulation benchmarks: RoboCasa-Kitchen (Nasiriany et al., 2024) and LIBERO (Liu et al., 2023a). We also demonstrate its applicability to real-world tasks using a 7-DoF manipulator. In Section 3.2, we further validate RS-CL in the setting where a VLA model is trained from scratch, starting from a pre-trained VLM. For an overview of the benchmark tasks and real-robot experiments, see Fig. 4.

**Implementation and training details.** We adopt GR00T N1.5 (GEAR, 2025) as our baseline VLA framework and, unless otherwise specified, we follow the training and inference settings from the original implementation. For the contrastive regularization path, the projection head $g_\psi$ is a 2-layer MLP with hidden dimension 2048 and projection dimension 128. The weighting coefficient $\lambda$ for $\mathcal{L}_{\text{RS-CL}}$ is initialized to 1.0 and decayed to 0 using a cosine schedule, such that representation refinement is emphasized in early training while accurate action prediction becomes the main focus later. For proprioceptive inputs, we primarily use the end-effector position $(x, y, z)$, 6D rotation, and gripper state. In the real-world tasks, we additionally explore the use of absolute joint positions of the 7-DoF manipulator to examine variations in proprioceptive configurations. Further training details for each experiment are provided in Appendix B.2.

**Baselines.** We primarily validate RS-CL on top of the GR00T N1.5 training pipeline, a state-of-the-art VLA model trained with large-scale robot trajectories. To provide context on the benchmarks, we also report the performance of representative VLA models, including $\pi_0$ (Black et al., 2025b), $\pi_0$-FAST (Pertsch et al., 2025), and GR00T N1 (Bjorck et al., 2025). For reproduced performance of $\pi_0$-FAST and $\pi_0$ on RoboCasa-Kitchen, we train for 30K and 60K gradient steps, respectively, with a global batch size of 64, following the original settings as closely as possible. In Section 3.2, we include as a baseline further-training the VLM with various instructions curated with robotics data, and then fine-tuning for action prediction. We make use of state-of-the-art embodied reasoning models such as RoboBrain (Team et al., 2025), VeBrain (Luo et al., 2025), and Cosmos-Reason1 (Azzolini et al., 2025), as well as models trained for discretized action prediction (Hung et al., 2025).

Table 1: **RoboCasa-Kitchen benchmark success rate (%).** Results include fine-tuned performance of representative VLA models ($\pi_0$-FAST, $\pi_0$, and GR00T N1). Performance of GR00T N1 is from the original work (Bjorck et al., 2025), while results of $\pi_0$, $\pi_0$-FAST, and GR00T N1.5 are reproduced. Best and runner-up results are highlighted in **bold** and underline, respectively.

| Method | 30 demos | | | 100 demos | | | 300 demos | | |
|---|---|---|---|---|---|---|---|---|---|
| | PnP | Others | Avg. | PnP | Others | Avg. | PnP | Others | Avg. |
| $\pi_0$ (Black et al., 2025b) | 20.0 | **61.3** | 47.8 | 32.7 | 71.6 | 58.7 | 45.0 | 72.9 | 62.5 |
| $\pi_0$-FAST (Pertsch et al., 2025) | 9.3 | 40.0 | 29.8 | 47.3 | 67.5 | 60.2 | 51.3 | 71.3 | 63.6 |
| GR00T N1 (Bjorck et al., 2025) | 0.4 | 25.9 | 17.4 | 2.2 | 47.0 | 32.1 | 22.6 | 63.1 | 49.6 |
| GR00T N1.5 (GEAR, 2025) | 30.8 | 56.9 | 48.2 | 51.8 | 70.0 | 63.9 | 55.3 | 70.9 | 65.7 |
| + RS-CL (Ours) | **41.5** | 58.8 | **53.0** | **58.0** | **71.8** | **67.2** | **59.8** | **74.6** | **69.7** |

Table 2: **LIBERO benchmark success rate (%).** Results include fine-tuned performance of representative VLA models ($\pi_0$-FAST, $\pi_0$, and GR00T N1). Performance of $\pi_0$-FAST, $\pi_0$ are from the original work (Black et al., 2025b; Pertsch et al., 2025), while the results of GR00T N1 and GR00T N1.5 are reproduced. Best results are highlighted in **bold**.

| Method | Spatial | Object | Goal | Long | Avg. |
|---|---|---|---|---|---|
| $\pi_0$ (Black et al., 2025b) | 96.4 | 98.8 | 95.8 | 85.2 | 94.1 |
| $\pi_0$-FAST (Pertsch et al., 2025) | 96.4 | 96.8 | 88.6 | 60.2 | 85.5 |
| GR00T N1 (Bjorck et al., 2025) | 95.6 | 97.6 | 94.2 | 89.6 | 94.3 |
| GR00T N1.5 (GEAR, 2025) | 98.2 | **99.4** | 97.2 | 87.8 | 95.7 |
| + RS-CL (Ours) | **98.4** | 98.6 | **98.2** | **90.4** | **96.4** |

## 3.1 FINE-TUNING EXPERIMENTS

We first evaluate RS-CL in a fine-tuning scenario, where it is integrated into a state-of-the-art pre-trained VLA model. This setup tests whether RS-CL can yield additional gains on weights already optimized for large-scale action prediction, demonstrating its ability to further enhance strong pretrained policies. We adopt RoboCasa-Kitchen (Nasiriany et al., 2024) and LIBERO (Liu et al., 2023a), two multitask benchmarks as our simulation experiments. To further validate the effectiveness of our method beyond simulation, we conduct real-robot experiments on a Franka Research 3 arm, covering both in-domain and generalization performance.

**Setup.** RoboCasa-Kitchen consists of 24 atomic manipulation tasks in a simulated kitchen environment with three camera views (2 exterior, 1 wrist camera). We evaluate RS-CL under varying numbers of demonstrations (30, 100, 300) using the publicly available dataset generated by MimicGen (Mandlekar et al., 2023). LIBERO is also a multitask simulation benchmark comprising four task suites: spatial, object, goal, and long (each with 10 tasks and 50 demonstrations per task), utilizing two camera views (1 exterior, 1 wrist camera). For LIBERO, we utilize the filtered dataset from Kim et al. (2024) and jointly train the four task suites (see Appendix B for details). To further assess whether RS-CL leads to more precise actions in task execution, we design our real-robot experiments primarily around pick-and-place tasks, which require accurate positioning during grasping and placing. We also introduce a challenging close-lid task, where the lid has a small handle that is more difficult to grasp than other objects. Once grasped, the wrist camera view becomes occluded, requiring placement to rely mainly using the exterior camera (see Fig. 4, right). We collect and train each method with 60 expert demonstrations for 4 pick-and-place tasks across diverse objects (teddy bear, sponge, cup, cube) and environments (box, bowl, plate, basket), and the close-lid task, utilizing two camera views (1 exterior, 1 wrist camera) (see Appendix C for details).

**Simulation results.** Table 1 summarizes the performance of RS-CL on RoboCasa-Kitchen. Across all dataset sizes, RS-CL consistently outperforms the original GR00T N1.5 fine-tuning framework. In particular, pick-and-place tasks exhibit a substantial improvement, with success rates rising from 30.3% to 41.5% (+11.2%). We attribute this gain to RS-CL's ability to generate more accurate actions during execution, which is particularly beneficial for pick-and-place tasks requiring precise positioning during grasping and placing. We further validate this in our following real-world experiments. RS-CL also improves performance on LIBERO (Table 2), confirming its robustness across different benchmarks.

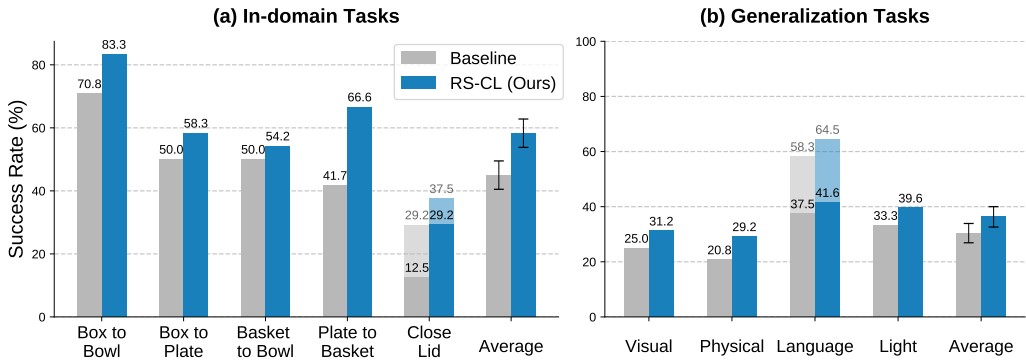

Figure 5: **Real-robot task success rate (%).** Results on **(a)** in-domain tasks (4 pick-and-place and 1 close-lid task), and **(b)** generalization tasks (visual, physical generalization, language grounding, and light variation). For the in-domain close-lid and language grounding tasks, we report both partial success (*e.g.*, successful pickup, language following; transparent bars) and full success (solid bars).

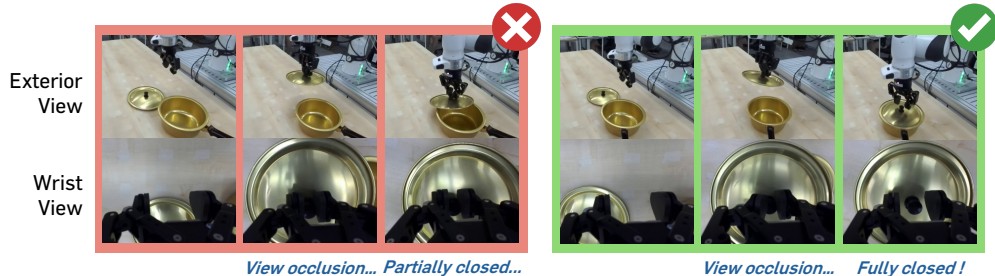

Figure 6: **Qualitative results on real-robot manipulation task.** Under partial-view occlusion at the wrist view, the baseline model (left) fails to align the lid with the pot, resulting in inaccurate placement. In contrast, RS-CL (right) achieves precise alignment and successful closing by effectively incorporating proprioceptive state information into its representation.

**Real-robot experiment results.** RS-CL consistently improves performance across real-robot tasks (see Fig. 5a). In particular, for the close-lid task, RS-CL brings improvements not only in partial success (*i.e.*, lifting the lid) but also larger gains in complete success (*i.e.*, accurately closing the pot) even under occluded viewpoints (see Fig. 6). We attribute this effect to two factors: (i) proprioceptive supervision enables more accurate positioning, and (ii) the proposed *view cutoff* augmentation promotes view-invariant representations, thereby improving robustness to partial occlusion. In addition, our generalization experiments show that RS-CL maintains strong generalization performance of VLAs across visual, physical shifts, and in the terms of language grounding (see Fig. 5b).

## 3.2 FROM-SCRATCH EXPERIMENTS

In this section, we evaluate the impact of RS-CL in a from-scratch training scenario, where we train a VLA model on top of general-purpose pre-trained VLM backbones of Qwen2.5-VL (Bai et al., 2025), GR00T N1.5 VLM (GEAR, 2025) and SigLIP2 (Tschannen et al., 2025). This setup directly aligns with our motivation that pre-trained VLM representations lack sensitivity to robotic signals, and allows us to validate whether explicitly aligning them to proprioceptive information yields performance gains. Furthermore, we compare the effect on RS-CL against baselines obtained by further training VLMs on robotics datasets.

**Setup.** We adopt RoboCasa-Kitchen as our main benchmark, and use 300 demonstrations for training all models. For the VLA training framework, we attach a randomly initialized action decoder to various pre-trained VLMs, with a lightweight adapter module $f_\phi$ in between. We freeze the VLM and train the adapter to refine condition representations, except for SigLIP2, where we experiment with an unfrozen VLM setting either to study how RS-CL interacts with different numbers of trainable backbone parameters. For the action decoder, we adopt a 16-layer DiT with 0.5B parameters. For the further-trained VLM baselines, we utilize RoboBrain (Team et al., 2025), VeBrain (Luo et al., 2025), and Cosmos-Reason1 (Azzolini et al., 2025), which are high-performing baselines further trained

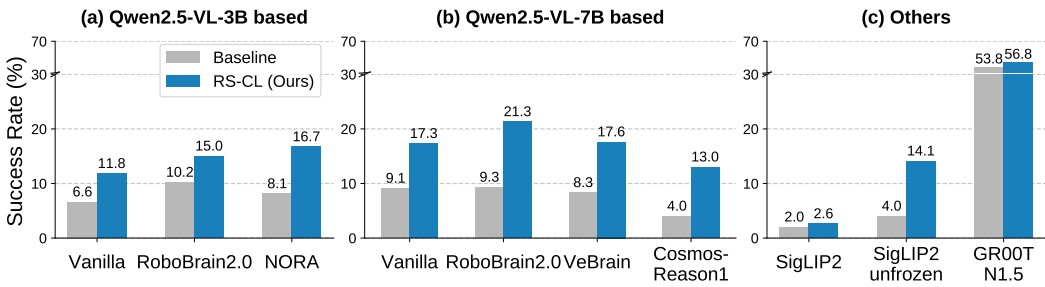

Figure 7: **From-scratch experiments.** Success rates (%) on RoboCasa-Kitchen for VLA models trained from various VLM backbones. Vanilla indicates Qwen2.5-VL. Results show the effects of RS-CL on top of backbones further trained with robotics data, based on **(a)** Qwen2.5-VL-3B, **(b)** 7B, as well as **(c)** SigLIP2 and GR00T N1.5 to provide diverse results across backbone and train capacity.

Table 3: **Ablation study.** Results report the average success rate (%) on RoboCasa-Kitchen with 300 demonstrations, analyzing the effect of **(a)** different distance definitions for soft-label supervision of robotic signals and **(b)** representation augmentation strategies for RS-CL.

| Soft-label target | Avg. |
|---|---|
| Baseline (*i.e.*, no regularization) | 65.7 |
| No soft label (*i.e.*, InfoNCE) | 67.3 |
| Next action sequence distance | 66.7 |
| Next single action distance | 66.8 |
| Current state distance | **69.7** |

(a) Soft-label target.

| Augmentation method | Avg. |
|---|---|
| No augmentation | 65.3 |
| Token cutoff | 66.3 |
| Feature cutoff | 67.5 |
| Span cutoff | 67.3 |
| View cutoff | **69.7** |

(b) Representation augmentation method.

from Qwen2.5-VL on embodied reasoning with robotics dataset, and NORA (Hung et al., 2025), which is trained on the Open-X-Embodiment (O'Neill et al., 2024) dataset to predict FAST (Pertsch et al., 2025) tokenized actions (see Appendix A.2 for details).

**Results on general-purpose VLM backbones.** Fig. 7 summarizes the effect of RS-CL when training VLA models from different pre-trained VLMs. Across all backbones, RS-CL consistently improves success rates, demonstrating that our representation regularization generalizes beyond a particular backbone model. On SigLIP2, RS-CL yields larger improvements from 4.0% to 14.1% when the backbone is unfrozen, indicating that RS-CL benefits from increased trainable capacity.

**Comparison to VLM training strategy.** Fig. 7 compares RS-CL with VLMs that are further trained on robotics datasets for tasks such as visual grounding, embodied reasoning, and discretized action prediction. While such further-trained VLMs, when used as conditioning models, provide only limited and often inconsistent gains across backbone families, RS-CL consistently delivers larger improvements. It achieves higher success rates than any of these adapted models on both Qwen2.5-VL-3B and 7B, and further enhances their benefits when combined with them. Even for GR00T N1.5, which is derived from Eagle 2.5 VLM (Chen et al., 2025) with enhanced grounding and reasoning capabilities, RS-CL provides additional gains. These results suggest that robotics-specific training alone may not fully close the gap between general-purpose VLM representations and the control signals required for action generation, while RS-CL effectively bridges much of this gap.

### 3.3 MORE ANALYSIS

**Effect of soft-label supervision target.** In Table 3a, we observe that standard InfoNCE improves over the baseline without contrastive learning, demonstrating the effectiveness of our training framework, namely contrastive representation regularization for VLA models (see Appendix D.1 for further analysis). However, alternative supervision signals (see Appendix B.3 for distance definition of targets) such as next action distances fall below vanilla InfoNCE. A plausible reason is that the next action itself serves as the prediction target, making it difficult to use as a reliable alignment signal. In contrast, the robot proprioceptive state provides a stable cue for representation alignment.

Table 4: **Comparison with temporal contrastive objectives.** Results report the average success rate (%) on RoboCasa-Kitchen with 30 demonstrations, together with FLOPs per sample in the forward process in training, and wall-clock training time. Best results among non-baseline methods are highlighted in **bold**.

| Method | Success rate (%, ↑) | FLOPs (×$10^{12}$, ↓) | Training time (hours, ↓) |
|---|---|---|---|
| Baseline | 48.2 | 2.58 | 23.06 (+ 0.0%) |
| Multi-view TCN | 50.0 | 7.53 | 47.77 (+107.1%) |
| Single-view TCN | 50.3 | 7.53 | 51.87 (+124.9%) |
| RS-CL | **53.0** | **2.91** | **23.49 (+ 1.3%)** |

**Effect of representation augmentation strategy.** In Table 3b, we observe limited improvements from similar representation-level cutoff operations (Shen et al., 2020), while our proposed view cutoff achieves the highest success rate. This shows that simulating viewpoint variation is particularly beneficial for robust representation learning in multi-view robotic manipulation settings. This is in line with prior works, addressing the effects of utilizing multi-view data for representation learning (Weinzaepfel et al., 2022; Seo et al., 2023).

**Quantitative analysis of representation alignment.** We further measure how RS-CL improves the alignment of VLM representations with robotic signals with CKNNA (Huh et al., 2024). As shown in Fig. 8, RS-CL increases representation similarity between learned embeddings and proprioceptive features, indicating that RS-CL successfully reshapes the embedding space toward capturing control-relevant signals. Details are described in Appendix B.3.

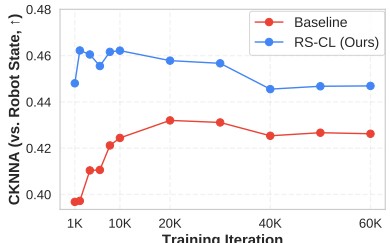

Figure 8: **Alignment to proprioceptive states.** We measure the alignment of condition representations inside trained VLA models, to the robot's proprioceptive states using CKNNA (Huh et al., 2024). RS-CL successfully improves the representation alignment to robot states of VLA models, compared to the model solely trained with action prediction loss.

**Comparison with temporal contrastive objectives.** To contextualize RS-CL among existing representation learning approaches, we compare against time-contrastive networks (TCN) (Sermanet et al., 2018), a widely used temporal contrastive method in robotics. TCN learns embeddings by enforcing that temporally close observations are mapped close together in representation space while observations from distant timesteps are pushed apart. We implement TCN as an auxiliary objective on top of GR00T-N1.5 and consider both a multi-view and a single-view variant for comparison. Details about the implementation and variants are described in Appendix B.3. Table 4 shows that both the multi-view and single-view TCN objectives slightly improve the success rate over the baseline, confirming that temporal contrastive regularization can strengthen the learned representations. However, these gains come at a substantial computational cost. The FLOPs per sample nearly triple ($2.58 \times 10^{12} \rightarrow 7.53 \times 10^{12}$), and the wall-clock training time more than doubles due to additional VLM forward passes for positive/negative pairs and the overhead of mining temporally structured samples. In contrast, RS-CL achieves the highest success rate of 53.3% while only modestly increasing FLOPs and wall-clock time (+1.3%), since the augmentation strategy, *view-cutoff* operates at the representation-level after a single VLM forward pass. Overall, RS-CL serves as an effective yet lightweight regularizer integrated into end-to-end VLA training, strengthening the conditioning representations without incurring significant additional computational overhead.

## 4 RELATED WORK

**Leveraging VLM representations for robot manipulation.** Vision-Language-Action (VLA) models have shown strong capabilities in robotic control by leveraging semantically enriched features from pre-trained Vision-Language Models (VLMs) (Zitkovich et al., 2023; Driess et al., 2023; Kim et al., 2024; Black et al., 2025b; Pertsch et al., 2025; Bjorck et al., 2025). A widely used architecture for VLA models consists of a pre-trained VLM and an action decoder with its parameters (Black

et al., 2025b; Bjorck et al., 2025; Shukor et al., 2025; Li et al., 2024; Zhou et al., 2025; Yang et al., 2025; Wen et al., 2025), training the VLM backbone with action prediction loss. Prior works have sought to further train VLMs for core knowledge of robot manipulation such as embodied reasoning and physical grounding (Ji et al., 2025; Luo et al., 2025; Azzolini et al., 2025; GEAR, 2025), or by discretized action prediction (Kim et al., 2025; Black et al., 2025a). Other methods jointly train the VLM with the action decoder on the aforementioned objectives. (Driess et al., 2025; Yang et al., 2025). Distinct from these approaches, our method does not rely on large-scale curated robotics datasets but instead improves VLM representations via a self-supervised objective.

**Contrastive representation learning.** Contrastive learning has been widely adopted for acquiring transferable representations from high-dimensional inputs (Oord et al., 2018; Chen et al., 2020; He et al., 2020; Laskin et al., 2020; Radford et al., 2021). In robotics, contrastive objectives have been applied to enable robust transfer of visuomotor policies, leveraging temporal consistency (Sermanet et al., 2018; Ma et al., 2023; Nair et al., 2022) or multi-view data (Seo et al., 2023). Recent efforts extend this idea to multimodal alignment (Rana et al., 2023; Lee et al., 2025; Myers et al., 2023), producing behaviorally grounded embeddings for control. While prior contrastive methods focus on training good representations for downstream tasks, we integrate contrastive learning into end-to-end VLA training, complementing the original action prediction objective.

## 5 CONCLUSION

In this work, we present *Robot State-aware Contrastive Loss (RS-CL)*, a simple and effective regularization method that explicitly aligns representations with robot proprioceptive states. Our experiments demonstrate that RS-CL consistently improves VLA performances, particularly on tasks requiring reliable and precise positioning. These findings highlight the importance of incorporating control-relevant structure into condition representations to enhance action prediction. We hope this work encourages further exploration of incorporating robot-centric signals, such as object pose or tactile feedback, to advance VLA models toward more precise and versatile robotic control.

## REPRODUCIBILITY STATEMENT

We provide our implementation details in Appendix A and further training and evaluation details in Section 3 and Appendix B.2 for reproducibility. Datasets for our benchmark experiments are publicly available, described at Appendix B.1.

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

# A HYPERPARAMETERS AND IMPLEMENTATION DETAILS

## A.1 HYPERPARAMETERS

For the weighting coefficient for $\mathcal{L}_{\text{RS-CL}}$, $\lambda$, we initialize to 1.0 and decayed to 0 using a cosine schedule by maximum training steps, such that representation refinement is emphasized in early training while accurate action prediction becomes the main focus later. For similarity temperature $\tau$ and soft weight temperature $\beta$, we use 0.2 and 1.0, respectively. We systematically analyze the sensitivity of our method to its main hyperparameters, and observe that performance remains stable over a wide range of settings, as summarized in Table 5.

## A.2 IMPLEMENTATION

DETAILS FOR FROM-SCRATCH VLA TRAINING

We attach a randomly initialized action decoder to various pre-trained VLMs, with a lightweight adapter module $f_\phi$ in between. Following GEAR (2025), we define $\text{VLM}(\mathbf{O}_t^V, \mathbf{c})$ as the hidden representation from layer 12 out of 36 layers for Qwen2.5-VL-3B variants and the GR00T N1.5 backbone. For Qwen2.5-VL-7B, we extract $\text{VLM}(\mathbf{O}_t^V, \mathbf{c})$ from layer 18 out of 28, which yields higher performance in our layer ablation study on LIBERO (see Table 6). For SigLIP, we instead use the final hidden representation as the condition embedding.

As the action decoder, we adopt a 16-layer DiT with 0.5B parameters. Empirically, we find that omitting a projection layer to reduce embedding dimensionality before conditioning improves performance (see Table 6). Accordingly, we do not apply such a layer. Instead, for Qwen2.5-VL-7B variants, we use a larger attention dimension that matches its hidden size $d_{\text{model}} = 3584$, while Qwen2.5-VL-3B uses $d_{\text{model}} = 2048$.

Table 5: Results report the average success rate (%) on RoboCasa-Kitchen with 30 demonstrations, analyzing the effect and sensitivity to main hyperparameters of RS-CL.

| Hyperparameters | Avg. |
|---|---|
| baseline | 48.2 |
| **$\lambda$ schedule** | |
| $\lambda$ decay $1.0 \rightarrow 0$ | **53.0** |
| $\lambda$ no schedule ($\lambda = 1.0$) | 50.7 |
| $\lambda$ no schedule ($\lambda = 0.5$) | 51.0 |
| **Similarity temperature $\tau$** | |
| $\tau = 0.01$ | 51.6 |
| $\tau = 0.02$ | **53.0** |
| $\tau = 0.05$ | 52.0 |
| $\tau = 0.1$ | 53.3 |
| $\tau = 1.0$ | 51.1 |
| **Distance temperature $\beta$** | |
| $\beta = 0.1$ | 51.2 |
| $\beta = 1.0$ | **53.0** |
| $\beta = 10.0$ | 49.8 |
| **Projection dimension** | |
| Proj. dim $2048 \rightarrow 64$ | 50.9 |
| Proj. dim $2048 \rightarrow 128$ | **53.0** |
| Proj. dim $2048 \rightarrow 256$ | 51.2 |
| **Batch size** | |
| baseline (bs32) | 48.4 |
| RS-CL (bs32) | 51.5 |
| baseline (bs64) | 48.2 |
| RS-CL (bs64) | **53.0** |
| **Training seeds (0, 7, 42)** | |
| baseline | 49.2 / 48.8 / 48.2 |
| RS-CL | 54.7 / 51.3 / **53.0** |

Table 6: **Hidden representation layer ablations on Qwen2.5-VL-7B backbone.** We report success rates (%) on the LIBERO benchmark, varying the hidden layer index used as the conditioning representation for VLA models trained from scratch.

| Layer | Spatial | Object | Goal | Long | Avg. |
|---|---|---|---|---|---|
| 12 (with projection) | 87.4 | 94.2 | 41.8 | 40.4 | 66.0 |
| 18 (with projection) | 86.8 | 83.4 | 61.6 | 44.0 | 69.0 |
| **18 (no projection)** | 85.2 | 89.4 | 73.2 | 36.2 | **71.0** |
| 24 (with projection) | 85.2 | 89.4 | 73.2 | 36.2 | 57.0 |

## A.3 HARDWARE DETAILS AND COMPUTATION OVERHEAD

All experiments are conducted on a single node equipped with $2 \times$ NVIDIA A100-SXM4-80GB GPUs and 64 CPU cores. Unless otherwise noted, we use a global batch size of 64 and train for 60K optimization steps.

To quantify the additional cost introduced by RS-CL, we measure both floating point operations (FLOPs) and wall-clock training time for our fine-tuning experiment in RoboCasa-Kitchen. Using a FLOPs profiler, we measure the forward FLOPs per sample during training. Table 7 summarizes the compute characteristics for both fine-tuning and from-scratch training.

Table 7: Compute overhead of RS-CL. We report estimated forward FLOPs per sample and total training time for 60K steps with global batch size 64.

| Method | FLOPs / sample (forward) | Training time (hrs) |
|---|---|---|
| Baseline | $2.576 \times 10^{12}$ | 23.06 |
| RS-CL | $2.909 \times 10^{12}$ | 23.49 |

The additional wall-clock cost introduced by RS-CL is negligible ($+1.25\%$), because the *view-cutoff* augmentation operates directly on the VLM embeddings and RS-CL only adds a lightweight projection head and soft contrastive loss on top of the backbone forward pass. In particular, it does not require extra forward passes through the VLM backbone or longer token sequences, so the dominant compute costs of training remain essentially unchanged.

## A.4 ALGORITHM

---

**Algorithm 1** Training VLA with Robot State-aware Contrastive Loss (RS-CL)

---

**Require:** Observations $\mathbf{O}_t^V$, instruction $\mathbf{c}$, robot state $\mathbf{q}$, ground-truth actions $\mathbf{A}_t$, hyperparameters $(\lambda, \beta, \tau)$

**Ensure:** Trained parameters $\theta, \phi, \psi$

1: **for** each training step **do**
2: $\quad \mathbf{h} \leftarrow f_\phi(\text{VLM}(\mathbf{O}_t^V, \mathbf{c}))$ $\qquad\qquad\qquad\qquad$ ▷ Encode inputs with frozen VLM + adapter
3: $\quad [\mathbf{h}, \mathbf{w}] \leftarrow f_\phi(\text{VLM}(\mathbf{O}_t^V, \mathbf{c}) \oplus \mathbf{u})$ $\qquad\qquad\quad$ ▷ Append summarization token
4: $\quad \mathbf{z} \leftarrow g_\psi(\mathbf{w})$ $\qquad\qquad\qquad\qquad\qquad\qquad$ ▷ Project summarization output
5: $\quad \tilde{\mathbf{z}} \leftarrow \texttt{ViewCutoff}(\mathbf{z})$ $\qquad\qquad$ ▷ View cutoff; Representation-level augmentation
6: $\quad \mathcal{L}_{\text{FM}} \leftarrow \|D_\theta(\mathbf{h}, \mathbf{A}_t^s, \mathbf{q}) - (\epsilon - \mathbf{A}_t)\|_2^2$ $\qquad\qquad\qquad$ ▷ Flow-matching loss
7: $\quad w_{ij} \leftarrow \frac{\exp(-\|\mathbf{q}_i - \mathbf{q}_j\|_2 / \beta)}{\sum_k \exp(-\|\mathbf{q}_i - \mathbf{q}_k\|_2 / \beta)}$ $\qquad\qquad$ ▷ Robot state-aware contrastive loss
8: $\quad \mathcal{L}_{\text{RS-CL}} \leftarrow -\sum_{i,j} w_{ij} \log \frac{\exp(\text{sim}(\mathbf{z}_i, \tilde{\mathbf{z}}_j)/\tau)}{\sum_k \exp(\text{sim}(\mathbf{z}_i, \tilde{\mathbf{z}}_k)/\tau)}$ $\qquad\qquad$ ▷ Contrastive loss
9: $\quad \mathcal{L} \leftarrow \mathcal{L}_{\text{FM}} + \lambda \mathcal{L}_{\text{RS-CL}}$ $\qquad\qquad\qquad\qquad$ ▷ Final joint objective
10: $\quad$ Update parameters $\theta, \phi, \psi$ via gradient descent

---

## B  SIMULATION EXPERIMENT DETAILS

### B.1  DATASET

For RoboCasa-Kitchen, we use the publicly available dataset [1] containing 3000 demonstrations generated with MimicGen (Mandlekar et al., 2023). For LIBERO, we use the publicly available dataset [2], consisting of all 270K samples from LIBERO-Spatial, LIBERO-Object, LIBERO-Goal, and LIBERO-Long, re-rendered by Kim et al. (2024).

### B.2  TRAINING AND EVALUATION DETAILS

For fine-tuning experiments on GR00T N1.5 (GEAR, 2025), we employ the publicly available pre-trained checkpoint [3]. We follow the original training and inference recipe of GEAR (2025), including the prior distribution $p(s) = \text{Beta}(\frac{a-s}{a}; 1.5, 1), a = 0.999$ for sampling the flow-matching timestep $s$ in equation 1. All models are trained with the *new_embodiment* tag. We omit the use of future tokens (Zheng et al., 2025), as they are beyond the scope of this work.

For RoboCasa-Kitchen, we train for 60K gradient steps with a global batch size of 64, using AdamW with a learning rate of 1e-4 under a cosine decay schedule and 3K warmup steps. For LIBERO, we adopt a smaller global batch size of 32, as this setting yields better performance in practice.

For $\pi_0$ and $\pi_0$-FAST, we use the pre-trained checkpoints [4] [5] to reproduce fine-tuned performance on RoboCasa-Kitchen. We train $\pi_0$ for 60K steps and $\pi_0$-FAST for 30K steps, both with a global batch size of 64. We set the learning rate to 2.5e-5 with cosine decay to 2.5e-6 and 1K warmup steps. At inference, we use an action horizon $H = 16$ and execute all actions without re-planning.

For RoboCasa-Kitchen, we evaluate all models with 1200 trials. For LIBERO, we evaluate 50 trials for each task, following Kim et al. (2024).

### B.3  ANALYSIS DETAILS

**Soft label target distance metric.**  For the ablation study on soft label targets in Sec. 3.3, we define distances as follows. For next single action and current state, we use Euclidean distance. For next action sequence, we use Dynamic Time Warping (DTW), which measures similarity between temporal sequences that may vary in speed. DTW requires an additional temperature hyperparameter $\gamma$, which we set to 10.0. The soft weight temperature $\beta$ and similarity temperature $\tau$ are fixed at 1.0 and 0.2, respectively.

**CKNNA measurement.**  CKNNA (Huh et al., 2024) is a nearest-neighbor variant of kernel alignment (Kornblith et al., 2019). We randomly sample 10 trajectories per task in RoboCasa-Kitchen, totaling 240 trajectories. Each trajectory is processed with a window size of 16, yielding 4415 transitions. We extract the embeddings from the adapter module $f_\phi$ (used as conditioning inputs to the action decoder) along with the corresponding proprioceptive states. We follow the implementation of Huh et al. (2024) and report results with $k = 10$, measuring the alignment between proprioceptive states and conditional representations in the VLA model.

**TCN implementation deatils.**  Since recent VLA models consumes multi-view observations (GEAR, 2025; Black et al., 2025b) in a single forward pass, the multi-view TCN variant samples negatives from timesteps outside a temporal margin range, while positives are generated by zeroing out a randomly selected camera view. The single-view TCN variant follows the original formulation, drawing positives from a nearby temporal window and negatives from a distant temporal window. Following the original work (Sermanet et al., 2018), we set the temporal margin for defining positive and negative pairs to 0.2s.

---

[1] https://huggingface.co/datasets/nvidia/PhysicalAI-Robotics-GR00T-X-Embodiment-Sim
[2] https://huggingface.co/datasets/physical-intelligence/libero
[3] https://huggingface.co/nvidia/GR00T-N1.5-3B
[4] gs://openpi-assets/checkpoints/pi0_base
[5] gs://openpi-assets/checkpoints/pi0_fast_base

## C   REAL WORLD EXPERIMENT DETAILS

### C.1   HARDWARE PLATFORM

We use Franka Research 3, a 7-DoF robotic arm equipped with a Robotiq 2F-85 gripper. For visual perception, we utilize the dual camera setup: a movable Stereolabs ZED 2 provides a global view, and a wrist-mounted ZED Mini captures a close-range view. Teleoperated demonstrations are collected using an Oculus Quest 2, and we log time-synchronized RGB images, joint states, and gripper width for training and evaluation. Demonstrations are recorded at 10 Hz.

### C.2   REAL-WORLD TASKS

The in-domain and generalization tasks (visual, physical generalization, and language grounding) along with their corresponding prompts and representative key frames from the real-world evaluation, are shown in Fig. 9– 12.

**In-domain tasks.**  We introduce four pick-and-place tasks (Box to Bowl, Box to Plate, Basket to Bowl, Plate to Basket), with varied objects (teddy bear, blue cube, blue cup, yellow sponge) for each task (see Fig. 9).

**Visual generalization.**  We use in-domain objects differing in color (*e.g.*, changing a blue cube to a green cube, or a yellow sponge to a blue sponge). We further introduce background variations by changing the tabletop covering or the target container (see Fig. 10).

**Physical generalization.**  We evaluate with unseen objects not used in training, including a yellow banana, purple grapes, red strawberry, and a yellow cup (different shape and texture from the blue cup used in training) (see Fig. 11).

**Language grounding.**  We place two in-domain objects at the pick up location, and specify which one to pick up (see Fig. 12).

**Light variation.**  We evaluate on in-domain tasks under significantly darker lighting conditions than those used in training (see Fig. 13).

### C.3   REAL-WORLD TRAINING AND EVALUATION DETAILS

**Dataset.**  We collect 60 demonstrations for each pick-and-place task and and for the close-lid task.

**Training.**  We jointly train a model with the 4 pick-and-place tasks, and another model for the close-lid task. For pick-and-place, we employ a cartesian action space with proprioceptive states, and for the close-lid task we use a joint action space to cover various configurations in manipulation.

**Evaluation.**  For real-robot evaluation, we report the average success rate over 24 trials for each pick-and-place task, with varied objects. In the close-lid task, outcomes are classified as full success (lid fully closed), partial success (partially closed), or failure (not closed). For physical generalization, we evaluate on unseen objects (yellow banana, purple grapes, red strawberry, yellow cup), with success defined as the accurate completion of the pick-and-place. We define language following as whether the gripper approaches the correct object, and task success as completing the instructed pick-and-place.

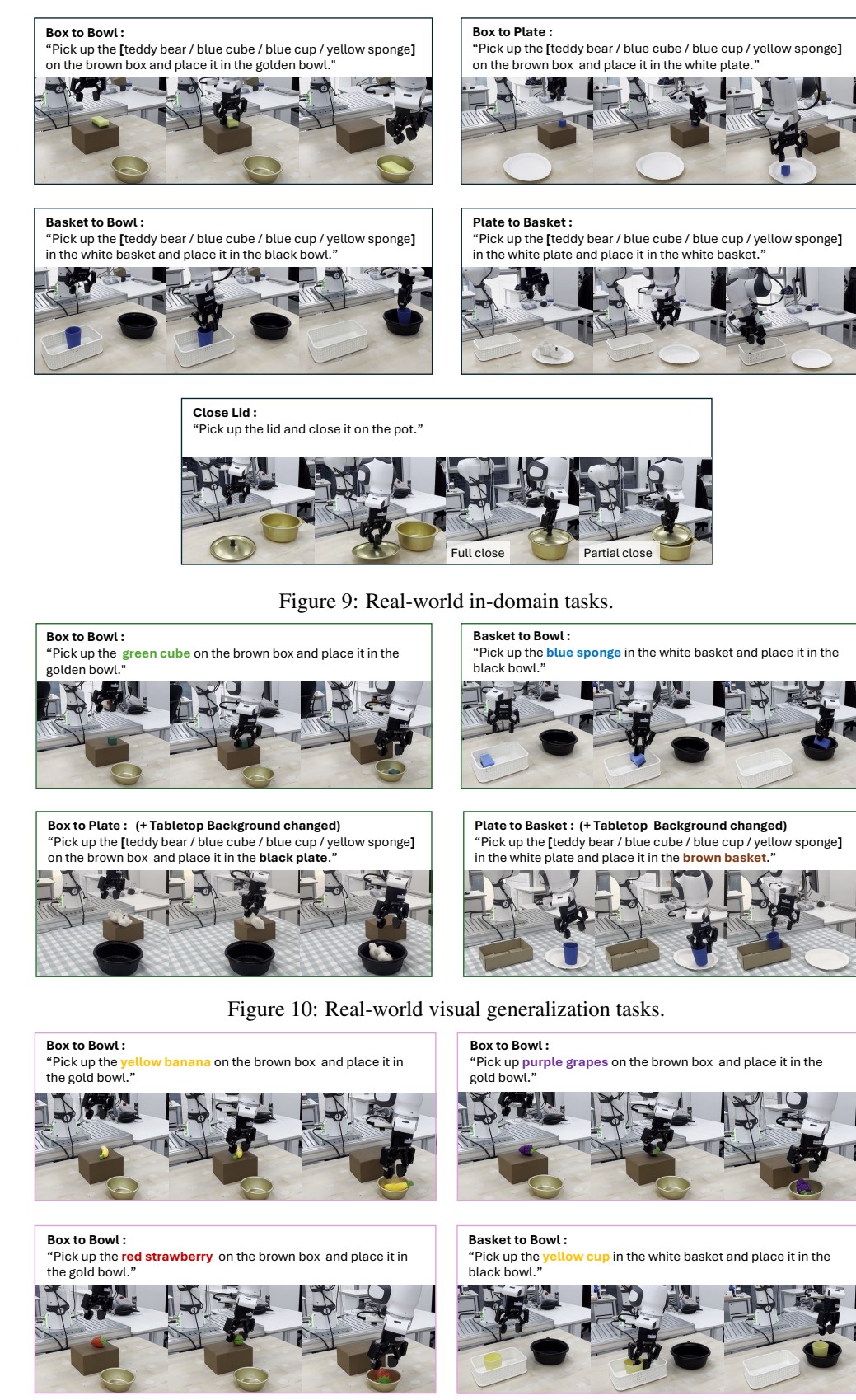

Figure 9: Real-world in-domain tasks.

Figure 10: Real-world visual generalization tasks.

Figure 11: Real-world physical generalization tasks.

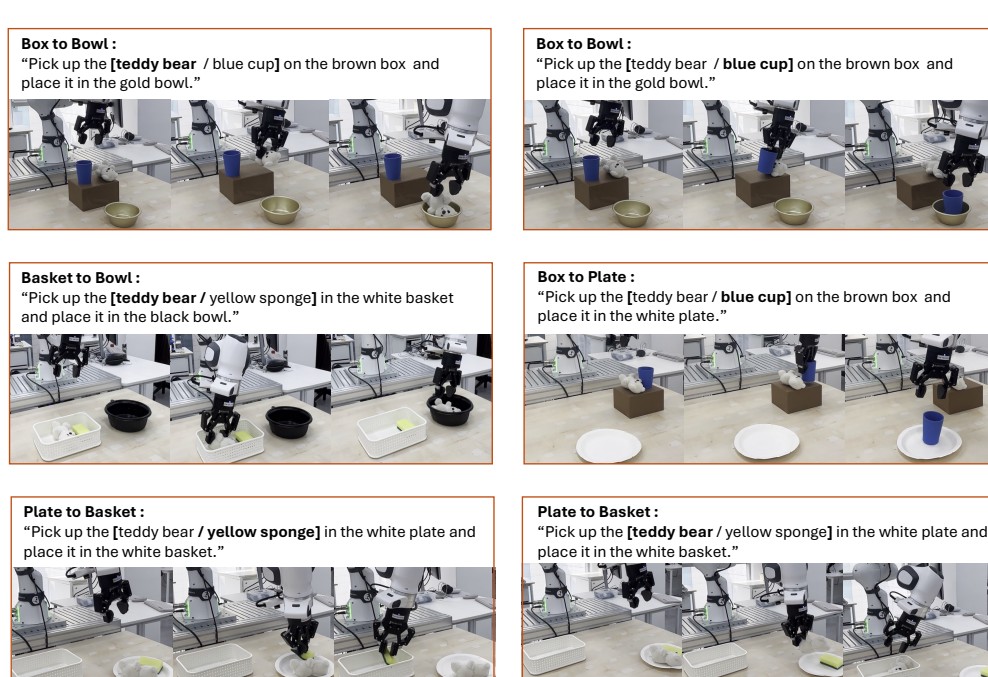

Figure 12: Real-world language grounding tasks.

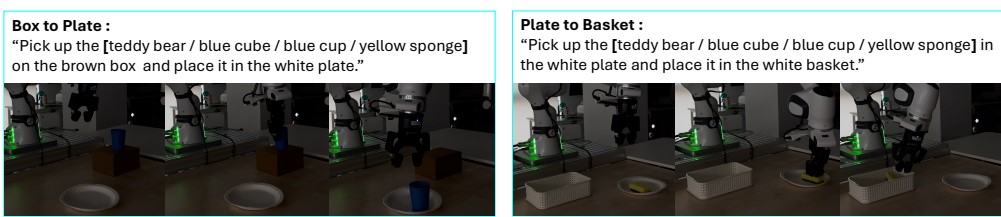

Figure 13: Real-world light variation tasks.

# D    FURTHER ANALYSIS

## D.1    CONTRASTIVE REPRESENTATION REGULARIZATION

Table 8: **RoboCasa-Kitchen benchmark success rate (%).**

| Method | 30 demos | | | 100 demos | | | 300 demos | | |
|---|---|---|---|---|---|---|---|---|---|
| | PnP | Others | Avg. | PnP | Others | Avg. | PnP | Others | Avg. |
| GR00T N1.5 (GEAR, 2025) | 30.8 | 56.9 | 48.2 | 51.8 | 70.0 | 63.9 | 55.3 | 70.9 | 65.7 |
| **+ CL (Ours)** | 36.0 | 55.0 | 48.1 | **59.3** | 69.0 | 65.0 | 57.0 | 72.6 | 67.3 |
| **+ RS-CL (Ours)** | **41.5** | **58.8** | **53.0** | 58.0 | **71.8** | **67.2** | **59.8** | **74.6** | **69.7** |

Table 9: **LIBERO benchmark success rate (%).**

| Method | Spatial | Object | Goal | Long | Avg. |
|---|---|---|---|---|---|
| GR00T N1.5 (GEAR, 2025) | 98.2 | **99.4** | 97.2 | 87.8 | 95.7 |
| **+ CL (Ours)** | 97.4 | 99.0 | 97.2 | 87.4 | 95.3 |
| **+ RS-CL (Ours)** | **98.4** | 98.6 | **98.2** | **90.4** | **96.4** |

On RoboCasa-Kitchen, a contrastive representation regularization, without other supervision from low-level robotic signals (*i.e.*, InfoNCE) improves the performance of GR00T N1.5 (**CL** at Table 8). This result indicates the effectiveness of our proposed training framework, together with the augmentation strategy *view cutoff*. With further supervision from the robot's proprioceptive states (**RS-CL** at

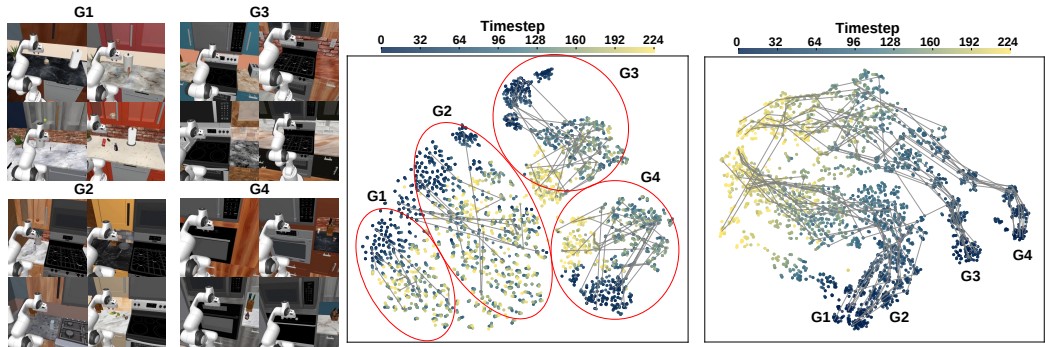

(a) Pre-trained VLM representations.     (b) RS-CL aligned representations.

Figure 14: **Detailed visualizations of VLM representations. (a)** We visualize VLM embeddings of robot episodes performing the same task "Open the microwave / cabinet door" across different scenes in RoboCasa-Kitchen. Pre-trained VLM representations form distinct clusters primarily based on visual appearance, and the task/timestep progress trajectories (*i.e.*, gray lines starting from blue to yellow dots) are not consistently aligned across these clusters. **(b)** With RS-CL, the representations still preserve scene-dependent grouping, but the task progress becomes geometrically aligned across groups (*i.e.*, from bottom toward the top-left region), indicating that episodes from different environments share a common progression direction in the embedding space.

Table 8), the performance further improves, highlighting the complementary benefit of incorporating proprioceptive information into VLM representations.

On LIBERO, CL performs comparably to the baseline (95.7% vs. 95.3%), but not improvements like RoboCasa-Kichen. This is likely due to the smaller batch size, where we train RoboCasa-Kitchen with a global batch size of 64, we train LIBERO with a global batch size of 32, for better performance of baseline GR00T N1.5 (bs64: 93.40 % vs. bs32: 95.65 %). This reduces the number samples calculated in the contrastive path, leading to lower improvement. However, with supervision of proprioceptive states (**RS-CL** at Table 8), the performance improves over baseline, despite the constraints.

## D.2 DETAILED VISUALIZATIONS OF VLM REPRESENTATIONS

We provide a more detailed explanation of Fig. 2 using the additional visualizations in Fig. 14. In the pre-trained VLM representation (Fig. 14a, we observe four prominent clusters (*i.e.*, G1–G4) that are clearly grouped by scene layout. Specifically, G1 corresponds to scenes where a flat, wide tabletop occupies most of the space in front of the robot; G2 to scenes with a tabletop in front and a stove or burner at the front right; G3 to scenes dominated by a stove or burner directly in front of the robot; and G4 to scenes where an oven is positioned in front of the robot. These clusters are thus primarily induced by the most visually dominant objects in the scene. However, the underlying task, "Open the microwave / cabinet door," mainly requires the robot arm to move upward and reach an overhead door in front of the robot, which is largely independent of these dominant background objects. Consistent with this mismatch, the timestep-indexed task progress trajectories (*i.e.*, gray lines from blue to yellow) do not align across clusters, indicating that the pre-trained VLM embeddings are organized by visual appearance rather than by control-relevant task progress.

In contrast, Fig. 14b shows the condition representations of the VLA model trained with RS-CL. While episode embeddings from different scenes still form partially separated groups, the common task progress becomes geometrically aligned across these groups. Trajectories consistently evolve from the bottom toward the top-left region of the embedding space. This suggests that RS-CL reshapes the conditioning representation to be more robot-state centric, so that episodes at similar task phases are aligned even when they come from visually distinct environments, while still preserving the semantic grouping from the pre-trained VLM.

Table 10: KNN task-phase classification accuracy (%) on task trajectories across scenes. RS-CL improves the representation's ability to classify task progress under visual changes and moves it closer to the performance obtained from ground-truth proprioceptive states.

| Features | Accuracy (%) |
|---|---|
| Pre-trained VLM representation | 1.2 |
| Trained only with action prediction loss | 20.3 |
| Trained with RS-CL | 22.9 |
| Ground-truth proprioceptive state | 25.6 |

### D.3 KNN TASK-PHASE CLASSIFICATION ANALYSIS

To further analyze how RS-CL shapes the conditioning representation, we perform a cross-scene KNN task-phase classification experiment on identical manipulation trajectories replayed in different visual environments. For each trajectory, we first align the executions across scenes using Dynamic Time Warping (DTW), and discretize the aligned time axis into 32 shared task-progress classes. Each timestep is assigned a scene-invariant phase label describing where it lies in the overall execution of the task.

Given a trained model, we extract the conditioning representation at each timestep and evaluate how well it encodes task progress under visual changes by training a KNN classifier in this representation space to predict the phase labels. We compare four feature choices: (i) the pre-trained VLM representation without any robot-action training, (ii) the representation from a model trained only with the action prediction loss (baseline), (iii) the representation from a model trained with RS-CL, and (iv) the ground-truth proprioceptive state, which serves as an upper-bound reference for phase information. We use a KNN classifier with $k = 5$.

Table 10 reports the resulting cross-scene KNN classification accuracy. The pre-trained VLM features achieve only 1.2% accuracy, indicating that task progress is essentially not recoverable under visual changes from the frozen VLM representation alone. Training with RS-CL boosts accuracy to 22.9%, nearly reaching the accuracy from ground-truth proprioceptive states (25.6%). This suggests that RS-CL encourages the conditioning representation to encode state observation more reliably in a way that is robust to changes in the visual background, providing a more stable signal for the downstream action decoder.

### D.4 ADAPTATION TO ANOTHER VLA ARCHITECTURE.

To verify that RS-CL remains effective regardless of the underlying action modeling paradigm of the baseline VLA model, we apply it to $\pi_0$-FAST (Pertsch et al., 2025), an autoregressive VLA model that predicts action tokens via next-token prediction instead of flow matching. Under the RoboCasa-Kitchen fine-tuning setting, $\pi_0$-FAST with RS-CL improves performance across all demonstration counts (see Table 11). These results suggest that, as long as the VLA model conditions its policy on a large-scale pre-trained VLM backbone, RS-CL is a broadly applicable and beneficial regularization strategy, independent of the specific action modeling design.

Table 11: **RoboCasa-Kitchen benchmark success rate (%).** Results include fine-tuned performance of $\pi_0$-FAST (Pertsch et al., 2025), an autoregressive VLA model and GR00T-N1.5 (GEAR, 2025), an flow-matching VLA model, with RS-CL. Best results within the same backbone indicated in **bold**.

| Method | 30 demos | 100 demos | 300 demos |
|---|---|---|---|
| $\pi_0$-FAST (Autoregressive) | 29.8 | 60.2 | 63.6 |
| + RS-CL (Ours) | **33.2** | **61.1** | **65.2** |
| GR00T-N1.5 (Flow-Matching) | 48.2 | 63.9 | 65.7 |
| + RS-CL (Ours) | **53.0** | **67.2** | **69.7** |

### D.5 MORE QUANTITATIVE RESULTS

We report further results of our RS-CL on a VLA trained from SigLIP2 (Tschannen et al., 2025), with varying number of demonstrations, and detailed results of our fine-tuning experiments in this section.

Table 12: **Detailed results on RoboCasa-Kitchen.** Task-wise success rates of GR00T N1.5 (GEAR, 2025) trained with, and without RS-CL, by different number of demonstrations.

| Task | GR00T N1.5 ($\mathcal{L}_{FM}$) | | | GR00T N1.5 ($\mathcal{L}_{FM} + \lambda \mathcal{L}_{RS\text{-}CL}$) | | |
|---|---|---|---|---|---|---|
| | 30 demos | 100 demos | 300 demos | 30 demos | 100 demos | 300 demos |
| RoboCasa Kitchen (24 tasks, PnP = Pick-and-Place) | | | | | | |
| Close Double Door | 44.0 | 86.0 | 80.0 | 54.0 | 78.0 | 86.0 |
| Close Drawer | 96.0 | 96.0 | 96.0 | 96.0 | 96.0 | 96.0 |
| Close Single Door | 98.0 | 94.0 | 98.0 | 88.0 | 98.0 | 98.0 |
| Coffee Press Button | 70.0 | 82.0 | 90.0 | 86.0 | 94.0 | 92.0 |
| Coffee Serve Mug | 64.0 | 72.0 | 58.0 | 74.0 | 66.0 | 70.0 |
| Coffee Setup Mug | 28.0 | 34.0 | 24.0 | 30.0 | 54.0 | 46.0 |
| Open Double Door | 80.0 | 92.0 | 82.0 | 72.0 | 80.0 | 84.0 |
| Open Drawer | 46.0 | 58.0 | 74.0 | 44.0 | 54.0 | 76.0 |
| Open Single Door | 64.0 | 58.0 | 78.0 | 66.0 | 60.0 | 74.0 |
| PnP from Cab $\rightarrow$ Counter | 28.0 | 42.0 | 54.0 | 38.0 | 54.0 | 60.0 |
| PnP from Counter $\rightarrow$ Cab | 36.0 | 54.0 | 54.0 | 40.0 | 58.0 | 68.0 |
| PnP from Counter $\rightarrow$ Microwave | 30.0 | 36.0 | 32.0 | 34.0 | 40.0 | 40.0 |
| PnP from Counter $\rightarrow$ Sink | 28.0 | 66.0 | 58.0 | 40.0 | 60.0 | 68.0 |
| PnP from Counter $\rightarrow$ Stove | 38.0 | 60.0 | 66.0 | 38.0 | 74.0 | 72.0 |
| PnP from Microwave $\rightarrow$ Counter | 24.0 | 44.0 | 50.0 | 46.0 | 50.0 | 48.0 |
| PnP from Sink $\rightarrow$ Counter | 40.0 | 52.0 | 60.0 | 54.0 | 62.0 | 68.0 |
| PnP from Stove $\rightarrow$ Counter | 22.0 | 60.0 | 68.0 | 42.0 | 66.0 | 54.0 |
| Turn Off Microwave | 62.0 | 86.0 | 94.0 | 62.0 | 84.0 | 94.0 |
| Turn Off Sink Faucet | 72.0 | 86.0 | 92.0 | 70.0 | 94.0 | 88.0 |
| Turn Off Stove | 10.0 | 14.0 | 28.0 | 10.0 | 8.0 | 28.0 |
| Turn On Microwave | 44.0 | 58.0 | 44.0 | 48.0 | 72.0 | 66.0 |
| Turn On Sink Faucet | 60.0 | 90.0 | 86.0 | 72.0 | 90.0 | 90.0 |
| Turn On Stove | 34.0 | 56.0 | 32.0 | 36.0 | 58.0 | 36.0 |
| Turn Sink Spout | 38.0 | 58.0 | 78.0 | 32.0 | 62.0 | 70.0 |
| **Average** | **48.2** | **63.9** | **65.7** | **53.0** | **67.2** | **69.7** |

Table 13: **Detailed results on RoboCasa-Kitchen.** Task-wise success rates (%) of reproduced $\pi_0$ (Black et al., 2025b) and $\pi_0$-FAST (Pertsch et al., 2025), by different number of demonstrations.

| Task | $\pi_0$ | | | $\pi_0$-FAST | | |
|---|---|---|---|---|---|---|
| | 30 demos | 100 demos | 300 demos | 30 demos | 100 demos | 300 demos |
| RoboCasa Kitchen (24 tasks, PnP = Pick-and-Place) | | | | | | |
| Close Double Door | 68.0 | 86.0 | 86.0 | 44.0 | 84.0 | 78.0 |
| Close Drawer | 94.0 | 94.0 | 96.0 | 84.0 | 96.0 | 94.0 |
| Close Single Door | 94.0 | 98.0 | 96.0 | 84.0 | 90.0 | 72.0 |
| Coffee Press Button | 66.0 | 80.0 | 88.0 | 20.0 | 82.0 | 90.0 |
| Coffee Serve Mug | 80.0 | 66.0 | 64.0 | 44.0 | 66.0 | 68.0 |
| Coffee Setup Mug | 20.0 | 32.0 | 38.0 | 2.0 | 34.0 | 38.0 |
| Open Double Door | 92.0 | 90.0 | 84.0 | 26.0 | 68.0 | 78.0 |
| Open Drawer | 44.0 | 56.0 | 62.0 | 36.0 | 58.0 | 68.0 |
| Open Single Door | 58.0 | 64.0 | 70.0 | 44.0 | 70.0 | 66.0 |
| PnP Cab $\rightarrow$ Counter | 14.0 | 22.0 | 18.0 | 12.0 | 22.0 | 30.0 |
| PnP Counter $\rightarrow$ Cab | 32.0 | 44.0 | 46.0 | 8.0 | 58.0 | 48.0 |
| PnP Counter $\rightarrow$ Microwave | 26.0 | 30.0 | 18.0 | 10.0 | 32.0 | 20.0 |
| PnP Counter $\rightarrow$ Sink | 32.0 | 44.0 | 58.0 | 2.0 | 46.0 | 56.0 |
| PnP Counter $\rightarrow$ Stove | 14.0 | 32.0 | 60.0 | 10.0 | 50.0 | 64.0 |
| PnP Microwave $\rightarrow$ Counter | 16.0 | 20.0 | 24.0 | 4.0 | 38.0 | 46.0 |
| PnP Sink $\rightarrow$ Counter | 22.0 | 24.0 | 66.0 | 12.0 | 56.0 | 62.0 |
| PnP Stove $\rightarrow$ Counter | 10.0 | 46.0 | 44.0 | 18.0 | 62.0 | 60.0 |
| Turn Off Microwave | 64.0 | 84.0 | 96.0 | 68.0 | 98.0 | 96.0 |
| Turn Off Sink Faucet | 72.0 | 86.0 | 94.0 | 48.0 | 76.0 | 94.0 |
| Turn Off Stove | 14.0 | 10.0 | 22.0 | 0.0 | 18.0 | 22.0 |
| Turn On Microwave | 58.0 | 82.0 | 70.0 | 52.0 | 68.0 | 88.0 |
| Turn On Sink Faucet | 80.0 | 82.0 | 86.0 | 40.0 | 66.0 | 74.0 |
| Turn On Stove | 26.0 | 68.0 | 42.0 | 12.0 | 52.0 | 38.0 |
| Turn Sink Spout | 50.0 | 68.0 | 72.0 | 36.0 | 54.0 | 76.0 |
| **Average** | **47.8** | **58.7** | **62.5** | **29.8** | **60.2** | **63.6** |

Table 14: **RoboCasa-Kitchen benchmark success rate (%)**. Employing SigLIP2 as our VLM backbone, we train a VLA model from scratch and report the average success rate by different number of demonstrations.

| Method | # of Demos | | |
|---|---|---|---|
| | 30 demos | 100 demos | 300 demos |
| SigLIP2 backbone VLA | 2.7 | 2.4 | 4.0 |
| + RS-CL (Ours) | **8.0** | **9.1** | **14.1** |

Table 15: **Detailed results of from-scratch experiments.** Task success rate (%) on the RoboCasa-Kitchen benchmark trained with 300 demonstrations. All models train a VLA from scratch, starting from each pre-trained VLM backbone. Best results within the same backbone indicated in **bold**.

| Backbone Model | Success Rate | | |
|---|---|---|---|
| | PnP | Others | Avg. |
| Qwen2.5-VL-3B (Bai et al., 2025) | 2.5 | 8.6 | 6.6 |
| + RS-CL (Ours) | **3.5** | **16.0** | **11.8** |
| NORA (Hung et al., 2025) | 1.5 | 11.4 | 8.1 |
| + RS-CL (Ours) | **3.5** | **23.3** | **16.7** |
| RoboBrain2.0-3B (Team et al., 2025) | 2.8 | 13.9 | 10.2 |
| + RS-CL (Ours) | **5.8** | **19.6** | **15.0** |
| Qwen2.5-VL-7B (Bai et al., 2025) | 2.5 | 12.4 | 9.1 |
| + RS-CL (Ours) | **9.8** | **21.1** | **17.3** |
| RoboBrain2.0-7B (Team et al., 2025) | 2.3 | 12.8 | 9.3 |
| + RS-CL (Ours) | **12.0** | **25.9** | **21.3** |
| VeBrain-7B (Luo et al., 2025) | 3.0 | 10.9 | 8.3 |
| + RS-CL (Ours) | **7.8** | **20.3** | **17.6** |
| Cosmos-Reason-7B (Azzolini et al., 2025) | 1.0 | 5.5 | 4.0 |
| + RS-CL (Ours) | **7.3** | **15.9** | **13.0** |
| SigLIP2 (Tschannen et al., 2025) | 0.3 | 2.9 | 2.0 |
| + RS-CL (Ours) | **0.8** | **3.5** | **2.6** |
| SigLIP2, unfrozen backbone | 3.3 | 4.4 | 4.0 |
| + RS-CL (Ours) | **17.3** | **12.5** | **14.1** |
| GR00T N1.5 VLM (GEAR, 2025) | 37.5 | 62.0 | 53.8 |
| + RS-CL (Ours) | **37.8** | **66.3** | **56.8** |

## E  DISCUSSION

**Limitations.** While RS-CL explicitly leverages proprioceptive states to align the representation space, it does not incorporate further signals in robotic manipulation, such as object poses or contact forces. These modalities often provide complementary information that is captured by robot's proprioception state. Extending RS-CL to integrate such modalities into the representations, represents an promising direction for future research.

**Future directions.** One promising extension is to apply RS-CL to settings with more complex proprioceptive spaces, such as humanoid robots or dexterous hand manipulation tasks. These domains involve high-dimensional and complex state representations, where aligning VLM embeddings with proprioceptive signals may be even more beneficial for accurate action prediction.

## F  USE OF LARGE LANGUAGE MODELS

Large language models were used to assist with drafting and polishing the writing of this paper.

