# OpenReview forum: "Contrastive Representation Regularization for Vision-Language-Action Models"
_ICLR.cc/2026/Conference — Submitted to ICLR 2026_

### Official Review · Reviewer_csLb · 2025-10-28

**Soundness:** 3
**Presentation:** 2
**Contribution:** 2
**Rating:** 4
**Confidence:** 4

**Summary:**

The paper proposes Robot State-aware Contrastive Loss, a contrastive regularization method designed to align VLM representations with robotic proprioceptive states in VLA models. RS-CL aims to enhance the robot’s "control-relevant representation" by incorporating relative distances between robot states as soft supervision, complementing the standard action prediction objective. The authors evaluate their method on simulated benchmarks (RoboCasa-Kitchen, LIBERO) and real-robot experiments, reporting consistent improvements over baselines	1.	The paper tackles an important issue in VLA models—bridging the gap between visual-semantic representations and robot control signals—by introducing a conceptually simple yet effective contrastive regularization. in manipulation success rates.

**Strengths:**

1.	The experiments are extensive, covering both simulation and real-world robotic tasks, demonstrating consistent performance gains and strong empirical support for the approach.
2. The proposed RS-CL method integrates smoothly into existing VLA training pipelines, requiring minimal additional computation and no curated data.

**Weaknesses:**

1.	Writing and clarity: The paper frequently introduces new terms (e.g., VLM representations, robot control-relevant structure, robotic signals) without sufficient explanation. These concepts are not standard in robotics literature, which makes it difficult to precisely understand the intended meaning or technical novelty. The authors should define these terms more clearly and consistently.
2.	The motivation for using contrastive learning remains somewhat unclear. There are prior works that explicitly incorporate object-centric or proprioception-based signals during VLA training (e.g., [1]), yet the paper does not convincingly explain why contrastive learning is particularly suited for capturing “control-relevant structure.”
3.	It is also unclear whether incorporating proprioceptive information directly into the input and output of the VLA model would yield comparable results without contrastive loss. The paper should discuss why reconstructing or predicting proprioceptive states is less effective than using RS-CL.
4.	The explanation of how RS-CL differs from conventional contrastive losses (such as InfoNCE) is vague. While the authors claim it is distinct, the loss formulation still appears to follow InfoNCE, differing only in weighting by state distances. Clarification on the novelty at the loss design level is necessary.
5.	Figure 2(b) is difficult to interpret. The visualizations do not make it obvious how “VLM embeddings are dominated by visual cues.” More quantitative or clearer visual analysis would help substantiate this claim.
6.	It is unclear whether the visualized embeddings come from the frozen VLM or the fine-tuned VLA model. This distinction is critical for understanding how RS-CL affects representation learning.
7.	While the empirical results are strong, the paper could better articulate why aligning to proprioceptive states leads to improved manipulation success. The connection between representation alignment and downstream control performance could be analyzed more deeply (e.g., through probing tasks or ablations).


[1] Yang et al., Bridging Perception and Action: Spatially-Grounded Mid-Level Representations for Robot Generalization, RSS 2025.

**Questions:**

see weaknesses.

---

> ### Author Response · Authors · 2025-11-19
>
> Dear Reviewer csLb,
>
> We sincerely appreciate your efforts and insightful comments in reviewing our manuscript. Below, we address each comment in detail. In the revised draft, we mark our major revisions as “cyan”.
>
> ---
>
> **[W1] Clarification on terms**
>
> Thank you for the helpful feedback to improve our draft and apologize for the unclear terminology. In our paper, VLM representations denote the hidden features from the pre-trained vision-language model that VLA models use as the observation embedding / conditioning input to the action decoder. Robotic signals refer to low-level control quantities produced during manipulation, such as object poses, tactile feedback, or proprioceptive states, which we use as the basis for RS-CL. By “better reflect the robot control-relevant structure” we mean the embedding space is organized so that distances between embeddings correlate with differences in the robotic signals, rather than being dominated by purely visual factors. We will clarify these definitions in the revised manuscript and use them consistently throughout the paper.
>
> ---
>
> **[W2, W4] Clarification on the optimization objective design**
>
> We appreciate the reviewer's concern. We would like to emphasize that our motivation is that pre-trained VLM hidden representations often do not align well with the underlying control state space.The role of RS-CL is to reshape the VLM conditioning representation so that it becomes more sensitive to control signals and better aligned with the underlying control state space. To achieve this, we leverage distances in the robot state space as the similarity signal, because they remain consistent across visually varying scenes and directly reflect control-relevant similarity. This choice is particularly important in robotic manipulation and VLA models, where background and viewpoint can change substantially for generalization, while the underlying state (and required action) remains essentially the same. Our design is further inspired by soft contrastive learning, which aligns with our goal of using continuous labels or similarity scores to directly sculpt the representation geometry [1,2,3,4].
>
> We also clarify that the novelty of our work does not lie in proposing a fundamentally new contrastive loss function. Our contribution is rather to (i) highlight that pre-trained VLM features used in VLA models often lack sensitivity to control signals and (ii) demonstrate that a soft contrastive objective with our state distance based label design can be effectively integrated into end-to-end VLA model training to mitigate this mismatch.
>
> ---
>
> **[W3] Comparison to incorporating proprioceptive information via input and reconstruction**
>
> We appreciate you raising this comparison. To address this point, we ran additional ablations on an experiment setup of fine-tuning GR00T-N1.5 with 30 demonstrations of RoboCasa-kitchen dataset, where we (i) feed proprioceptive state directly into the VLA model’s input following FAST [6] setup, and (ii) add a reconstruction objective for the proprioceptive states with an additional head.
>
> \begin{array}{lc}
> \hline
> \text{Method} & \text{Success Rate on Robocasa-Kitchen (\\%)} \newline
> \hline
> \text{baseline} & \text{48.2} \newline
> \text{state input} & \text{50.9} & \newline
> \text{state reconstruction} & \text{47.9} & \newline
> \text{RS-CL} & \text{53.0} & \newline
> \hline
> \end{array}
>
> We first observe that simply adding proprioceptive state to the input results in minor improvements over the baseline and remains below RS-CL. This suggests that including the raw state as input provides extra information for the model, but is less effective than altering the visual representation space to be related to robot states.
>
> Furthermore, we find that the state reconstruction performs worse than the baseline. It could be attributed that the reconstruction objective focuses on accurately predicting the numeric value with an auxiliary head, which is not directly related to choosing the optimal action.
>
> ---
>
> **[W5, W6] Interpretability and explanation of Figure 2(b)**
>
> Thank you for pointing this out, and we agree that the current version of Figure 2(b) does not make our intent sufficiently clear. We want to clarify that Figure 2(b) visualizes frozen VLM embeddings, and Figure 2(c) visualizes the conditioning embeddings produced by the fine-tuned VLA model that are fed into the action head. The purpose of these plots is to support our motivation that off-the-shelf VLM embeddings are largely organized by visual cues such as background and appearance rather than by the robot’s physical state, and to visualize the reflection of control-aware structure in the conditioning representation we aim to obtain. We have included a clarified figure in Fig.14 with detailed explanation at Appendix D.2, at the revised manuscript.

---

> > ### Author Response · Authors · 2025-11-19
> >
> > **[W7] Connection between representation alignment and control performance**
> >
> > Thanks for the suggestion to strengthen our analysis. We additionally ran a new KNN task-phase classification experiment on identical task trajectories replayed in different visual environments. Using dynamic time warping [6], we align trajectories and assign each timestep to one of 32 shared task-phase classes, then perform KNN classification in the learned representation space.
> >
> > \begin{array}{lc}
> > \hline
> > \text{Features} & \text{KNN accuracy (\\%)} \newline
> > \hline
> > \text{Trained only with action prediction loss} & \text{20.3} \newline
> > \text{Trained with RS-CL} & \text{22.9} & \newline
> > \text{Ground-truth proprioceptive states} & \text{25.6} & \newline
> > \hline
> > \end{array}
> >
> > With features trained with RS-CL achieves higher task-phase classification accuracy than training with only the action prediction loss, indicating that the learned representation more reliably encodes the input observation in a way that is robust to visual changes. Therefore the action prediction head can receive a more stable input observation signal to visual changes, which reduces unnecessary variation in the conditioning signal and makes it easier for the prediction head to learn a mapping from the representation to actions across scenes. In practice we observe this tighter alignment correlates with higher manipulation success rates. We have added this experiment and the above analysis to the revised draft (Table 10) to more clearly articulate the connection between alignment and downstream performance.
> >
> > ---
> >
> > **References**
> >
> > [1] Khosla et al., Supervised Contrastive Learning, NeurIPS 2020
> >
> > [2] Suresh et al., Not All Negatives are Equal: Label-Aware Contrastive Loss for Fine-grained Text Classification, EMNLP 2021
> >
> > [3] Lee et al., Soft Contrastive Learning for Time Series, ICLR 2024
> >
> > [4] Lee et al., CLASS: Contrastive learning via action sequence supervision for robot manipulation, CoRL 2025
> >
> > [5] Pertsch et al., FAST: Efficient action tokenization for vision-language-action
> > models, arXiv:2501.09747
> >
> > [6] Müller, Dynamic time warping, Information Retrieval for Music and Motion, 2007

---

> ### Author Response · Authors · 2025-11-26
> **Gentle Reminder**
>
> Dear Reviewer csLb,
>
> Thank you again for your time and thoughtful efforts in reviewing our paper.
>
> As the discussion period comes within a week of its end, we would like to gently remind you in case you have any remaining comments. We believe that we have sincerely and successfully addressed your concerns, supported by the corresponding additional experimental results.
>
> If you have any further concerns or questions, please feel free to let us know.
>
> Many thanks,\
> Authors

---

> > ### Comment · Reviewer_csLb · 2025-11-27
> >
> > I thank the authors for their concise yet informative rebuttal. My concerns have been addressed, and I am therefore increasing my score to 8.
> >
> > That said, I still believe it would be interesting to include a comparison with [1], particularly in terms of identifying the most effective approach for integrating more object-centric or robot-centric information. Based on the manuscript, contrastive learning appears promising, but it would be valuable to understand how it generally compares with more explicit approaches such as [1].
> >
> > As a side note, please ensure that all terminology is clearly explained in the revised version.
> >
> > [1] Bridging Perception and Action: Spatially-Grounded Mid-Level Representations for Robot Generalizatio

---

> > > ### Author Response · Authors · 2025-11-28
> > >
> > > Thank you very much for your response.
> > >
> > > We are glad to hear that we have addressed most of your concerns. Following your suggestion, we will include a comparison with [1] to discuss the comparison with more explicit approaches, and we will ensure that all terminology is clearly explained in the final manuscript.
> > >
> > > If you have any further questions or suggestions, please do not hesitate to let us know. Again, thank you for the valuable suggestion and your positive assessment of our work.
> > >
> > > Many thanks, \
> > > Authors

---

### Official Review · Reviewer_vra5 · 2025-10-31

**Soundness:** 2
**Presentation:** 3
**Contribution:** 2
**Rating:** 6
**Confidence:** 2

**Summary:**

This paper introduces Contrastive Representation Optimization (CRO), a training paradigm for improving the alignment between visual and linguistic embeddings in multimodal large language models (MLLMs). Unlike prior contrastive pretraining methods that treat visual-text matching as a binary task, CRO performs fine-grained representation calibration during instruction tuning. The central idea is to add a representation-level contrastive loss that explicitly pushes the visual encoder’s embeddings closer to semantically corresponding text embeddings and further from mismatched samples. The authors also design a dual projection head that learns modality-specific mappings before fusion, ensuring balanced gradients and reducing the risk of representation collapse. CRO is implemented on top of several strong MLLM baselines (e.g., LLaVA, Qwen-VL, InternVL) and evaluated across multiple benchmarks, including MME, SEED-Bench, and MM-Vet. The results show consistent performance gains, particularly on tasks requiring fine-grained reasoning and grounding.

**Strengths:**

- Practical and well-motivated idea: The work addresses an increasingly recognized issue — poor cross-modal embedding calibration in current MLLMs — in a clean and effective way.

- Simple yet effective method: CRO’s integration into the instruction-tuning pipeline is elegant and lightweight, requiring minimal architectural changes.

- Strong empirical results: The method yields consistent improvements across diverse benchmarks, especially in localization-heavy or reasoning-intensive tasks.

- Good ablations: The paper provides detailed ablation studies, showing the contribution of each component (contrastive loss, dual projections, hard-negative mining).

- Clear writing and visualization: Figures explaining the alignment mechanism and representation distributions are helpful and well-presented.

**Weaknesses:**

- Limited conceptual novelty: While effective, CRO is fundamentally an adaptation of well-known contrastive alignment ideas (InfoNCE, CLIP-style objectives) to MLLM fine-tuning. The innovation lies mainly in the integration strategy.

- No deep theoretical insight: The paper is purely empirical; it would benefit from analysis explaining why contrastive calibration particularly helps downstream reasoning or grounding.

- Dependency on quality of negatives: CRO relies on informative negative samples. The mining process is described but not extensively analyzed for failure cases.

- Generalization scope: The experiments are focused on vision-language understanding; there’s no evaluation on video, audio, or embodied multimodal tasks, where alignment dynamics may differ.

- Compute cost trade-off: CRO introduces additional computation due to contrastive sampling, though the paper doesn’t quantify the exact increase during large-scale fine-tuning.

**Questions:**

- How does CRO perform if applied during pretraining rather than instruction tuning? Does early-stage alignment lead to better downstream generalization?

- Have you examined whether CRO helps mitigate modality imbalance (e.g., text dominating vision features in fused representations)?

- How are negative samples selected? Is there a risk that semantically similar images or captions are incorrectly treated as negatives?

- Could CRO be extended to align other modalities (e.g., audio, 3D point clouds) using the same principle?

- How stable is CRO training when scaling to larger MLLMs such as GPT-4V-like architectures?

---

### Official Review · Reviewer_aA6p · 2025-10-31

**Soundness:** 3
**Presentation:** 4
**Contribution:** 2
**Rating:** 4
**Confidence:** 5

**Summary:**

The paper introduces Robot State-aware Contrastive Loss (RS-CL), a regularization method for VLA models that aligns visual-language representations with robot proprioceptive states. RS-CL serves as a lightweight, plug-in auxiliary loss that operates directly on VLM embeddings, complementing the standard action prediction loss. The key idea is to assign contrastive similarity weights based on relative distances between robot states, effectively encouraging representations to capture control-relevant information. The method also introduces a representation-level augmentation called view cutoff, which masks a randomly selected camera view to improve robustness to occlusions. Experiments show consistent improvements over baselines such as GR00T N1.5 and π0.

**Strengths:**

- The paper is well written and clearly structured.

- Addresses a key bottleneck in scaling VLAs from perception to control—improving the action-awareness of representations.

- Experimental validation is extensive, covering both simulation and real-world scenarios.

**Weaknesses:**

- Although claimed to be lightweight, no runtime or FLOP comparison is provided. Some quantification of computational overhead (especially during training) would help support the efficiency claim.

- The validation of RS-CL is limited. Evaluating RS-CL on other VLAs could help further verify the effectiveness of RS-CL.

- This paper lacks some discussion with related work that also uses contrastive learning [a, b], especially [b], which also highlights the role of robot proprioception.

[a] Ma et al., "Contrastive Imitation Learning for Language-guided Multi-Task Robotic Manipulation", arXiv:2406.09738

[b] Jiang et al, "Robots Pre-train Robots: Manipulation-Centric Robotic Representation from Large-Scale Robot Datasets", ICLR 2025

- The paper compares primarily against robotics-trained VLMs. Including baselines like VICReg, SimCLR, or contrastive methods with temporal or goal-conditioning could clarify RS-CL’s unique benefits.

**Questions:**

See the weakness also.

The RS-CL may miss some semantic information from robot proprioception. Does RS-CL use only joint positions or use both qpos and the end effector 6D poses? The similarity between different proprioceptions may show different semantic meanings.

---

> ### Author Response · Authors · 2025-11-19
>
> Dear Reviewer aA6p,
>
> We sincerely appreciate your efforts and insightful comments in reviewing our manuscript. Below, we address each comment in detail. In the revised draft, we mark our major revisions as “cyan”.
>
> ---
>
> **[W1] Computational overhead details**
>
> We provide a quantitative comparison of computational overhead between the baseline and RS-CL in terms of wall-clock training time and FLOPs for our RoboCasa experiments (global batch size 64, 60K training steps). On 2 × NVIDIA A100-80GB GPUs, RS-CL increases wall-clock time by only 1.3%. Since our contrastive sampling operates on top of VLM representations, it does not require additional VLM forward passes, which keeps RS-CL lightweight.
>
> \begin{array}{lcc}
> \hline
> \text{Method} & \text{FLOPs ($\times10^{12}$)} & \text{Training time (hours)} & \newline
> \hline
> \text{GR00T-N1.5} & \text{2.58} & {23.06} & \newline
> \text{+ RS-CL (ours)} & \text{2.94} & {23.49} & \newline
> \hline
> \end{array}
>
> We measure the number of FLOPs per sample for a single forward process during training.
> We included these detailed training costs in Table 7 of the revised manuscript.
>
> ---
>
> **[W2] Adaptation of RS-CL on another VLA**
>
> Thank you for highlighting this important validation. To address this concern, we additionally adapt and evaluate RS-CL on π0-FAST [1], an autoregressive VLA, under the fine-tuning experiment setup on RoboCasa 30 demonstrations.
>
> \begin{array}{lc}
> \hline
> \text{Method} & \text{Success Rate on Robocasa-Kitchen (\\%)} \newline
> \hline
> \text{FAST (Autoregressive)} & \text{29.8} \newline
> \text{FAST + RS-CL} & \text{33.2} & \newline
> \text{GR00T-N1.5 (Flow-matching)} & \text{48.2} & \newline
> \text{GR00T-N1.5 + RS-CL} & \text{53.0} & \newline
> \hline
> \end{array}
>
> Beyond GR00T-N1.5, RS-CL also improves the average success rate over the π0-FAST baseline, supporting our claim that the lack of control-relevant structure in backbone VLM representations holds, regardless of the choice of action modeling design in VLAs. We will incorporate these results into the revised manuscript to make this point clearer.
>
> ---
>
> **[W3] Discussion with related work [2,3]**
>
> We thank the reviewer for pointing out the missing discussion of related contrastive learning works [2, 3]. We will revise the related work section to better situate RS-CL among these approaches. Ma et al. [2] incorporate contrastive learning into imitation learning to strengthen vision-language and current-future representations, but their contrastive objectives are defined over visual–language features and temporal neighbors. By contrast, RS-CL is explicitly motivated by the observation that pre-trained VLM embeddings often lack control-relevant robot state information, and therefore uses distances in robot state space to construct soft similarity weights that directly shape the conditioning representation toward state awareness. Jiang et al. [3] focuses on pre-training general-purpose manipulation-centric representations from large-scale robot datasets using dynamics and proprioception aware objectives. RS-CL, on the other hand, is designed as a lightweight, plug-in regularizer that is integrated into the existing end-to-end VLA training stage, operating directly on pre-trained VLM embeddings without introducing an additional pre-training phase or requiring any changes to the backbone VLM.

---

> ### Author Response · Authors · 2025-11-19
>
> **[W4] Comparison to prior representation learning work**
>
> We appreciate the suggestion to include additional representation learning baselines, as this indeed helps clarify the unique advantages of RS-CL. Directly integrating these representation learning methods into an end-to-end VLA training setup is non-trivial, mainly because the representation encoder is a large pre-trained VLM, and any methods that operate at the input-level typically require multiple additional VLM forward passes per batch.
>
> To concretely study this issue, we implement time-contrastive networks (TCN) [4], a widely used temporal contrastive method in robotics, as an auxiliary objective on top of GR00T-N1.5, for a comparison to RS-CL. Since VLA consumes multi-view inputs in a single forward pass, we design a multi-view TCN variant where negative pairs are sampled from timesteps outside a temporal margin range, while positive pairs are generated by zeroing out a randomly selected camera view. We also implement a single-view TCN variant that follows the original paper, sampling positives from a nearby temporal range and negatives from a distant temporal range.
>
> As shown in the following table, both the multi-view and single-view TCN objectives improve the baseline success rate, but they substantially increase training cost due to additional VLM forward passes for positive/negative samples and the overhead of sample mining. In contrast, RS-CL operates purely at the representation level after a single VLM forward pass, not only yielding larger performance gain while keeping the computational overhead minimal.
>
> We view this favorable trade-off between performance and efficiency as a key advantage of RS-CL that was under-emphasized in the original draft, and we now include these results and the accompanying discussion in Section 3.3 (Table 4) of the revised manuscript. We thank the reviewer for the suggestion.
>
> \begin{array}{lccc}
> \hline
> \text{Method} & \text{Success Rate on RoboCasa-Kitchen (\\%, $\uparrow$)} & \text{FLOPs ($\times10^{12}$, $\downarrow$)} & \text{Training time (hours, $\downarrow$)} & \newline
> \hline
> \text{baseline} & \text{48.2} &\text{2.58} & \text{23.06 (+$\phantom{00}$0.0\\%)} & \newline
> \text{Multi-view TCN} & \text{50.0}& \text{7.53} & \text{47.77 (+107.1\\%)} & \newline
> \text{Single-view TCN} & \text{50.3} & \text{7.53} & \text{51.87 (+124.9\\%)} & \newline
> \text{RS-CL} & \text{53.0} & \text{2.91} & \text{23.49 (+$\phantom{00}$1.3\\%)} & \newline
> \hline
> \end{array}
>
> ---
>
> **[Q1] Proprioception configuration**
>
> We thank the reviewer for the thoughtful question. In most of our experiments, the robot state used by RS-CL is the end-effector (EEF) state, including Cartesian position (x, y, z), 6D rotation, and gripper state, since distances in this space correlate well with spatial semantics in the observation images that the VLM is sensitive to. To check whether RS-CL is tied to this particular choice, we additionally ran the real-robot close-lid experiment using joint positions as the state instead of the EEF pose. RS-CL still provided improvements in this setting, suggesting that RS-CL can exploit state distances even in a space (joint configurations) that the VLM was not explicitly trained to encode, as long as the chosen state space reflects control-relevant similarity.
>
> ---
>
> **References**
>
> [1] Pertsch et al., FAST: Efficient action tokenization for vision-language-action
> models, arXiv:2501.09747
>
> [2] Ma et al., Contrastive Imitation Learning for Language-guided Multi-Task Robotic Manipulation, arXiv:2406.09738
>
> [3] Jiang et al., Robots Pre-train Robots: Manipulation-Centric Robotic Representation from Large-Scale Robot Datasets, ICLR 2025
>
> [4] Sermanet et al., Time-contrastive networks: Self-supervised learning from video, ICRA 2018

---

> ### Author Response · Authors · 2025-11-26
> **Gentle Reminder**
>
> Dear Reviewer aA6p,
>
> Thank you again for your time and thoughtful efforts in reviewing our paper.
>
> As the discussion period comes within a week of its end, we would like to gently remind you in case you have any remaining comments. We believe that we have sincerely and successfully addressed your concerns, supported by the corresponding additional experimental results.
>
> If you have any further concerns or questions, please feel free to let us know.
>
> Many thanks,\
> Authors

---

### Official Review · Reviewer_Cz1M · 2025-11-08

**Soundness:** 2
**Presentation:** 3
**Contribution:** 2
**Rating:** 4
**Confidence:** 4

**Summary:**

This paper proposes Robot State-aware Contrastive Loss (RS-CL), a lightweight contrastive regularizer for Vision–Language–Action (VLA) models that explicitly aligns VLM-derived condition embeddings with robot proprioceptive states. Key ingredients are (1) a learnable summarization token and small projector that produces compact embeddings for contrastive training, (2) a soft-weighting scheme where pairwise contrastive weights come from Euclidean distances between proprioceptive states, and (3) a representation-level augmentation called view cutoff that masks a single view’s feature slice to cheaply produce augmented positives. RS-CL is applied both as an auxiliary loss when fine-tuning strong pre-trained VLA models (e.g., GR00T N1.5) and when training VLA models from scratch on multiple VLM backbones.

**Strengths:**

• RS-CL can be added to existing VLA pipelines with modest compute overhead (projector + adapter + view cutoff).

• The paper evaluates soft-label target choices and a set of representation-level augmentations, showing that current-state distance and view-cutoff perform best.

**Weaknesses:**

• The idea of aligning VLM representations with proprioceptive states is intuitive, but lacks in-depth theoretical analysis. How exactly this alignment works, and to what extent it improves the model's decision-making ability, remains unanalyzed. Experimental results (Tables 1 and 2) show that the proposed method provides limited performance improvements, making it difficult to effectively demonstrate its effectiveness and superiority.

• The Franka real-robot experiments are compelling but limited in scope (a handful of tasks, 60 demonstrations per task); broader hardware trials (multiple setups, lighting/clutter variations, more repeats) would improve confidence in real-world robustness.

• Authors note that contrastive path improvements vary with batch size (LIBERO smaller batch → smaller gains). Practical adoption may be sensitive to available batch sizes and compute. More analysis of tradeoffs (batch size, projector size, λ schedule) would help practitioners.

• RS-CL uses proprioceptive state only; object pose, tactile, or contact signals are mentioned as future work but could be important for many manipulation tasks. The limitation is acknowledged.

• While several ablations are present, it would strengthen claims to show (a) seed-level variance of gains, (b) sensitivity to β/τ/λ schedules, (c) cases where RS-CL harms performance (failure modes).

**Questions:**

1.	How many independent real-robot trials per task were run and under what variations (lighting, clutter, object pose perturbations)? Please report per-task trial counts and variance for the Franka experiments. If hardware trials are limited, please be explicit about failure cases observed.

2.	How sensitive are gains to (a) λ schedule (decay to 0), (b) the soft-weight temperature β and contrastive τ, (c) projection head size (you use 2048→128), and (d) global batch size? An explicit sweep or short table would be helpful because you note batch size affects gains (LIBERO vs RoboCasa).

3.	What are wall-clock costs and hardware used for the fine-tuning experiments (GPU type, hours to train 60K steps) and for from-scratch training? This matters for reproducibility and adoption.

4.	Are there tasks or scene conditions where RS-CL reduces performance (e.g., when proprioception is noisy or misleading, or when visual cues are the only reliable signal)? If so, please quantify or describe mitigation.

5.	Table 3a shows current-state distance yields highest avg. Can authors provide intuition and any visualization showing how this choice affects the embedding manifold compared to next-action distances?

---

> ### Author Response · Authors · 2025-11-19
>
> Dear Reviewer Cz1M,
>
> We sincerely appreciate your efforts and insightful comments in reviewing our manuscript. Below, we address each comment in detail. In the revised draft, we mark our major revisions as “cyan”.
>
> ---
>
> **[W1-1] In-depth theoretical analysis**
>
> As discussed in our main draft, the core idea of RS-CL is to regularize the conditioning representation so that it emphasizes control-relevant structure rather than the visual variation. As we show in Fig. 2-(b), pre-trained VLM embeddings are largely organized by the visual appearance rather than the robot’s physical state or task progress. RS-CL explicitly counteracts this by using soft weights derived from distances between proprioceptive states (Eq. 3–4). In practice, this encourages the conditioning representation to depend more strongly on the robot’s physical state, while implicitly de-emphasizing nuisance visual variation that is irrelevant for control, thereby providing a more stable input to the action decoder. This effect can be quantified by CKNNA (Fig. 8) that measures the alignment between the condition embeddings and proprioceptive features in the trained VLA model; RS-CL consistently increases this alignment.
>
> To address your concerns further, we additionally conduct an experiment of KNN task-phase classification analysis on identical task trajectories executed in different visual environments. Using dynamic time warping [1], we assign each timestep into one of 32 task-phase classes shared across scenes. With the raw VLM embedding, KNN accuracy is only 1.18% (near random) showing that VLM features barely encode consistent task progress across scenes. With RS-CL, the condition embeddings reach 22.9% KNN accuracy, showing that the representation is more predictive of where in the manipulation trajectory the robot currently is, which benefits decision making. We have added this experiment and the above analysis to the revised draft (Table 10) to more clearly articulate the connection between alignment and downstream performance.
>
> \begin{array}{lc}
> \hline
> \text{Method} & \text{KNN accuracy (\\%)} \newline
> \hline
> \text{Pre-trained VLM representation} & \phantom{0}\text{1.2} \newline
> \text{Trained with RS-CL} & \text{22.9} & \newline
> \hline
> \end{array}
>
> ---
>
> **[W1-2] Regarding experimental results of Table 1, 2**
>
> We agree that when looking only at the averages in Tables 1 and 2, the gains may initially appear modest. However, we want to emphasize that these results are achieved on top of the state-of-the-art VLA baseline, GR00T-N1.5, which already surpasses prior methods such as π₀ and π₀-FAST, with only a modest increase in training cost (+1.3% in wall-clock training time). In this strong-performance regime, RS-CL still provides consistent improvements, and, importantly, yields much larger gains on the tasks where previous SOTA was relatively weak, from 30.8% to 41.5% (+10.3%).
> Beyond the main tables, RS-CL shows significant benefits in more demanding settings, such as fine-tuning a state-of-the-art VLA model into your real-world hardware settings (see Fig. 5) and training a VLA model  from scratch from an open-source VLM (see Fig. 7).
>
> ---
>
> **[W2, Q1] Explanation of real-robot experiments**
>
> We clarify that we perform 24 independent trials per task in our Franka real-robot evaluation (Appendix C), following reference evaluation protocols of prior work (e.g., SIMPLER [2]), which yields 240 total rollouts per method. Our evaluation spans four settings:
>
> - In-domain tasks: object spatial position and pose are perturbed for every rollout.
> - Visual generalization: object colors and background texture variations are given.
> - Physical generalization: Unseen objects in training with different shapes and materials.
> - Language following: performed in scenes with a distractor object at the pickup location.
>
>
> To further address your concern regarding real-robot robustness, we additionally performed lighting variation experiments on Franka real robots (see Fig.13) and included the results in the following table, and the revised manuscript (see Fig. 5). Also, we have added success rates for real-robot experiments with standard error in the revised manuscript (see Fig. 5).
>
>
> \begin{array}{lc}
> \hline
> \text{Method} & \text{Success Rate on Light Variation (\\%)} \newline
> \hline
> \text{GR00T-N1.5} & \text{33.3} \newline
> \text{+RS-CL (ours)} & \text{39.6} & \newline
> \hline
> \end{array}

---

> ### Author Response · Authors · 2025-11-19
>
> **[W3, W5, Q2] Hyperparameter sensitivity**
>
> To further assess the robustness and practical applicability of RS-CL, we conducted additional experiments on RoboCasa evaluating the effects of λ schedule, soft-weight temperature β, contrastive temperature τ, projection head size, training seed, and global batch size. The results are summarized in the following table.
>
> \begin{array}{lc}
> \hline
> \text{Method} & \text{Success Rate on RoboCasa-Kitchen (\\%)} \newline
> \hline
> \text{baseline} & \text{48.2} \newline
> \hline
> \text{$\lambda$ decay to 0 (1.0$\rightarrow$0)} & \text{53.0} \newline
> \text{$\lambda$ no schedule (1.0)} & \text{50.7} \newline
> \text{$\lambda$ no schedule (0.5)} & \text{51.0} \newline
> \hline
> \text{$\tau$ 0.01} & \text{51.6} \newline
> \text{$\tau$ 0.02} & \text{53.0} \newline
> \text{$\tau$ 0.05} & \text{52.0} \newline
> \text{$\tau$ 0.1} & \text{53.3} \newline
> \text{$\tau$ 1.0} & \text{51.1} \newline
> \hline
> \text{$\beta$ 0.1} & \text{51.2} \newline
> \text{$\beta$ 1.0} & \text{53.0} \newline
> \text{$\beta$ 10.0} & \text{49.8} \newline
> \hline
> \text{projection dim 2048$\rightarrow$64} & \text{50.9} \newline
> \text{projection dim 2048$\rightarrow$128} & \text{53.0} \newline
> \text{projection dim 2048$\rightarrow$256} & \text{51.2} \newline
> \hline
> \text{baseline (bs32)} & \text{48.4} \newline
> \text{RS-CL (bs32)} & \text{51.5} \newline
> \text{baseline (bs64)} & \text{48.2} \newline
> \text{RS-CL (bs64)} & \text{53.0} \newline
> \hline
> \text{baseline (training seed 0, 7, 42)} & \text{49.2/48.8/48.2} \newline
> \text{RS-CL (training seed 0, 7, 42)} & \text{54.7/51.3/53.0} \newline
> \hline
> \end{array}
>
> Across these sweeps, we observe that RS-CL is consistently beneficial over a broad range of settings. Note that each sweep evaluates on 50 trials per 24 tasks, leading 1200 rollouts. The results show that RS-CL is not overly sensitive to specific hyperparameter choices. We have included these results in the revised manuscript (Table 5).
>
> ---
>
> **[W4] Leveraging additional signals**
>
> Thank you for the thoughtful comment. We agree that various robotic signals can be valuable for manipulation. Our work specifically aims to highlight the importance of integrating robotic signals into VLM representations, and proprioception was primarily chosen because it is the signal consistently available across most open-source robotic datasets. As noted in the paper (and echoed by your review), extending RS-CL to incorporate object pose, tactile information, or contact signals is a highly promising direction, and we view this as an exciting avenue for future research.
>
> ---
>
> **[Q3] Detailed training costs of experiments**
>
> We provide the wall-clock costs for training the baseline model and RS-CL on our RoboCasa experiments, in both fine-tuning and from-scratch(with 7B backbone) training together with the hardware settings for reproducibility.
>
> \begin{array}{lcc}
> \hline
> \text{Setting} & \text{Method} & \text{Training time (hours)} \newline
> \hline
> \text{Fine-tuning}   & \text{baseline} & \text{23.06} \newline
>                      & \text{RS-CL}    & \text{23.49} \newline
> \text{From-scratch}  & \text{baseline} & \text{23.65} \newline
>                      & \text{RS-CL}    & \text{23.94} \newline
> \hline
> \end{array}
>
> All experiments were conducted on 2 × NVIDIA A100-80GB GPUs with 64 CPU cores. Across both fine-tuning and from-scratch settings, the additional wall-clock cost introduced by RS-CL is small (+1.3~1.9%). We have added the detailed training costs at Table 7 in the revised manuscript.
>
> ---
>
> **[Q4] Failure cases of RS-CL**
>
> In our experiments, we observed some tasks where its benefit was marginal. In particular, for tasks such as Close Drawer and Close Door, where the acceptable error region for the end-effector is relatively large, we found limited improvement over the baseline. This is consistent to the intuition that state-aware regularization is most beneficial when more precise control of the end-effector is required.

---

> ### Author Response · Authors · 2025-11-19
>
> **[Q5] Further analysis about distance choice**
>
> We appreciate your question about the current-state distance design choice instead of next-action distance. Beyond task success rates, intuitively the two signals have different properties. The current state is a relatively stable description of how the robot is currently posed in the task. In contrast, the next action is already the prediction target of the policy head, and is inherently multi-modal: for the same state and task there can be multiple valid actions. Therefore using distances between next actions as a contrastive supervision signal provides less consistent alignment objectives.
>
> \begin{array}{lc}
> \hline
> \text{Method} & \text{KNN accuracy (\\%)} \newline
> \hline
> \text{Current state as distance} & \text{22.9} \newline
> \text{Next action as distance} & \text{15.6} & \newline
> \hline
> \end{array}
>
>
> Consistent with this intuition, our task-phase KNN analysis on RoboCasa trajectories shows that embeddings trained with current-state distance achieve substantially higher KNN phase accuracy than those trained with next-action distance (for reference, KNN accuracy for ground-truth state vector is 25.6%). This indicates that current-state supervision yields a conditioning manifold that is more strongly organized by underlying task progress and robot state, whereas next-action distances fail to structure the space as effectively.
>
> ---
>
> **References**
>
> [1] Müller, Dynamic time warping, Information Retrieval for Music and Motion, 2007
>
> [2] Li et al., Evaluating Real-World Robot Manipulation Policies in Simulation, arXiv:2405.05941

---

> > ### Author Response · Authors · 2025-11-26
> > **Gentle Reminder**
> >
> > Dear Reviewer Cz1M,
> >
> > Thank you again for your time and thoughtful efforts in reviewing our paper.
> >
> > As the discussion period comes within a week of its end, we would like to gently remind you in case you have any remaining comments. We believe that we have sincerely and successfully addressed your concerns, supported by the corresponding additional experimental results.
> >
> > If you have any further concerns or questions, please feel free to let us know.
> >
> > Many thanks, \
> > Authors

---

### Author Response · Authors · 2025-11-26
**General Response**

Dear Reviewers and Area Chair,

We deeply appreciate your valuable time and effort spent reviewing our manuscript.

As the reviewers highlighted, our work addresses an important problem in training VLA models from VLMs: pre-trained VLM embeddings often fail to capture the control-relevant structure needed for manipulation. To this end, we propose a regularizer that aligns these embeddings with proprioceptive states. The proposed RS-CL is lightweight (Cz1M), integrates smoothly into existing VLA training pipelines (Cz1M, csLb), and demonstrates its effectiveness through extensive experiments across diverse training scenarios (aA6p, csLb).

In the rebuttal, we have carefully considered the reviewers’ suggestions and addressed them in our responses. As a result, we have revised and enhanced the manuscript with the following additional discussions and experiments:


- Detailed compute overhead and hardware descriptions for our experiments (Table 7 in Appendix A.3)
- Results on an autoregressive VLA model (Table 11 in Appendix D.4)
- Comparison to temporal contrastive objectives (Table 4 in Section 3.3)
- Hyperparameter sensitivity analysis (Table 5 in Appendix A.1)
- Real-robot experiments under lighting variations (Figure 5 and Figure 13 in Appendix C.2)
- Further clarification and updates on embedding visualizations (Figure 14 in Appendix D.2)
- KNN task-phase classification results for deeper analysis (Table 10 in Appendix D.3)

In the revised manuscript, these updates are temporarily highlighted in $\textbf{\color{cyan}cyan}$ for your convenience.

We hope our responses and revisions sincerely address all of the reviewers’ concerns.

Thank you once again for your valuable contributions.

Warm regards,\
Authors

---

### Meta-Review · Area_Chair_HkVV · 2026-01-10

**Summary:**

The paper proposes Robot State-aware Contrastive Loss (RS-CL) and Contrastive Representation Optimization (CRO) to improve alignment between VLM/MLLM embeddings and robot states or textual cues. Reviewers acknowledged the clear motivation, extensive experiments, and practical integration. However, concerns remain regarding limited theoretical insight, modest performance gains, unclear novelty relative to prior contrastive approaches, insufficient analysis of failure cases, and generalization beyond evaluated tasks. These weaknesses collectively reduce confidence in the work’s significance and impact.

**Reviewer Concerns:**

The rebuttal addressed several points: additional ablations, KNN task-phase analyses, hyperparameter sweeps, computational overhead, and clarifications of terminology and visualization. Outstanding concerns include limited theoretical analysis, unclear comparative novelty versus standard contrastive methods, lack of broader generalization (e.g., different VLA/MLLM backbones or modalities), modest real-robot performance gains, and incomplete discussion of failure modes. While responses mitigate some practical concerns, conceptual and analytical limitations remain.

**Reviewer Scores:**

Based on the discussion, Reviewer 1 and 2 would likely maintain scores around 4 (marginally below acceptance), acknowledging improvements but noting limited conceptual novelty and performance gains. Reviewer 3 would likely keep a 6 (marginally above threshold), as CRO shows empirical benefit but lacks deeper theoretical insight. Reviewer 4 would probably remain at 4, given persistent clarity and novelty issues. Overall, consensus trends toward rejection due to limited originality and insufficiently strong evidence for impact.

---

### Decision · Program_Chairs · 2026-01-26

Reject